# Prostate Cancer: Genetics, Epigenetics and the Need for Immunological Biomarkers

**DOI:** 10.3390/ijms241612797

**Published:** 2023-08-14

**Authors:** Guzel Rafikova, Irina Gilyazova, Kadriia Enikeeva, Valentin Pavlov, Julia Kzhyshkowska

**Affiliations:** 1Institute of Urology and Clinical Oncology, Bashkir State Medical University, 450077 Ufa, Russia; rafikovaguzel@gmail.com (G.R.); gilyasova_irina@mail.ru (I.G.); kienikeeva@bashgmu.ru (K.E.); pavlov@bashgmu.ru (V.P.); 2Institute of Biochemistry and Genetics, Ufa Federal Research Center of the Russian Academy of Sciences, 450054 Ufa, Russia; 3Laboratory for Translational Cellular and Molecular Biomedicine, Tomsk State University, 634050 Tomsk, Russia; 4Genetic Technology Laboratory, Siberian State Medical University, 634050 Tomsk, Russia; 5Institute of Transfusion Medicine and Immunology, Mannheim Institute of Innate Immunosciences (MI3), Medical Faculty Mannheim, Heidelberg University, 68167 Mannheim, Germany; 6German Red Cross Blood Service Baden-Württemberg—Hessen, 68167 Mannheim, Germany

**Keywords:** cancer genetics, cancer epigenetic, tumor microenvironment, monocytes, TAMs, prostate cancer, cancer biomarkers

## Abstract

Epidemiological data highlight prostate cancer as a significant global health issue, with high incidence and substantial impact on patients’ quality of life. The prevalence of this disease is associated with various factors, including age, heredity, and race. Recent research in prostate cancer genetics has identified several genetic variants that may be associated with an increased risk of developing the disease. However, despite the significance of these findings, genetic markers for prostate cancer are not currently utilized in clinical practice as reliable indicators of the disease. In addition to genetics, epigenetic alterations also play a crucial role in prostate cancer development. Aberrant DNA methylation, changes in chromatin structure, and microRNA (miRNA) expression are major epigenetic events that influence oncogenesis. Existing markers for prostate cancer, such as prostate-specific antigen (PSA), have limitations in terms of sensitivity and specificity. The cost of testing, follow-up procedures, and treatment for false-positive results and overdiagnosis contributes to the overall healthcare expenditure. Improving the effectiveness of prostate cancer diagnosis and prognosis requires either narrowing the risk group by identifying new genetic factors or enhancing the sensitivity and specificity of existing markers. Immunological biomarkers (both circulating and intra-tumoral), including markers of immune response and immune dysfunction, represent a potentially useful area of research for enhancing the diagnosis and prognosis of prostate cancer. Our review emphasizes the need for developing novel immunological biomarkers to improve the diagnosis, prognosis, and management of prostate cancer. We highlight the most recent achievements in the identification of biomarkers provided by circulating monocytes and tumor-associated macrophages (TAMs). We highlight that monocyte-derived and TAM-derived biomarkers can enable to establish the missing links between genetic predisposition, hormonal metabolism and immune responses in prostate cancer.

## 1. Epidemiology

The World Health Organization (WHO) estimates that cancer was responsible for 10 million deaths in 2020, accounting for nearly one in six deaths. There were 1,414,259 new cases of prostate cancer (PC) worldwide and 375,304 deaths from this type of cancer in 2020 according to the Global Cancer Observatory (GCO) (gco.iarc.fr, accessed on 10 March 2023). PC ranks third in terms of incidence and second in terms of cancers affecting men globally [1]. In terms of mortality, PC ranks eighth worldwide (Table 1). Overall, PC statistics are relatively similar across different regions, although it ranks first among cancers affecting men in Latin America and eighth in Asia [2]. This suggests the potential impact of racial, ethnic, socioeconomic factors, and variations in the biology of prostate carcinogenesis, including the genetic predisposition of certain groups to developing biologically aggressive forms of the disease. Even when considering geographically diverse populations with similar access to healthcare, men of African descent have a higher incidence of PC and worse prognosis. African American men, Caribbean men, and black men in Europe exhibit higher incidence rates than white men, with mortality rates approximately twice as high, indicating a genetic predisposition to PC development [3,4]. Analyses of PC biopsies from men in sub-Saharan Africa have revealed a significant proportion of high-grade tumors (Gleason score 8+). However, variations in healthcare access and differences in cancer case registration across countries limit the reliability of conclusions regarding the correlation between race, ethnicity, geography, and PC aggressiveness based solely on epidemiological data [3]. Furthermore, the significance of racial differences in PC outcomes diminishes significantly if the male population in the analyzed countries has equal access to healthcare services [4,5,6].

The worldwide incidence and mortality of PC are strongly correlated with increasing age, with the average age at the time of primary diagnosis being 67 years [7]. Epidemiological studies conducted globally have reported the highest incidence of PC in individuals aged 75–79 years. In 2022, the incidence rates were recorded as 155 cases per 100,000 for ages 55–59, 510 cases per 100,000 for ages 65–69, and 751 cases per 100,000 for ages 75–79 [2].

Nineteen international studies, spanning from 1935 to 2014 and including countries such as Scandinavia, Caucasus, France, Hungary, Greece, Japan, China, Singapore, and North America, examined the histopathological evidence of PC in autopsies of 6024 males aged between 50–59, regardless of the cause of death. These studies found that approximately half of the individuals had histopathological evidence of prostate cancer, although only 3.8% died from PC [8].

The prognosis for patients newly diagnosed with PC varies significantly depending on the stage of the disease. If the disease is detected at early stages, men with localized PC can have a life expectancy of up to 99% for more than 10 years, depending on age, comorbidities, and chosen treatment strategy [9]. International statistics indicate that around 5% of men diagnosed with PC develop distant metastases, and the overall five-year survival rate for metastatic disease is 30% [9]. Almost all patients with metastatic PC eventually progress to a castration-resistant form that is resistant to androgen deprivation therapy. These factors contribute significantly to the morbidity and mortality associated with PC [9].

While it is widely recognized that the immune system plays a crucial role in controlling cancer initiation and progression, there has been insufficient attention given to epidemiological studies examining the link between immune status, subclinical inflammation of infectious or endogenous origin, and the incidence of prostate cancer. It would be of great interest to further investigate how inflammation, leading to subclinical activation of innate immunity, may facilitate the development of prostate cancer.

## 2. Genetic Predisposition for Prostate Cancer

### 2.1. Genetic Markers of PC

The vast majority of PC cases are sporadic and caused by turning off tumor suppressor genes and turning on oncogenes, and only 5–10% of PC cases have family history, being caused by germline mutations [10]. The presence of a first-degree PC relative is a risk factor associated with a two- to three-fold risk of PC compared to men without a family history of PC [11]. Epidemiological and case–control studies have shown the inheritance of specific mutations in PC susceptibility genes and reported that patients with these mutations have an increased risk of the disease [9]. Thus, *BRCA* 1/2 germline mutations in men increase the risk of PC relative to the average population level by 3–8 times. Genetic alterations in *BRCA2* in PC patients are associated with aggressive behavior of the disease [11]. The sequencing of DNA samples in men with a family PC history reveals a high frequency of DNA repair gene mutations, such as *BRCA1/2*, *HOX*, *ATM*, *CHECK2, PALB2* and *RAD51D*, *RNase L* (*HPC1*, lq22), *MSR1* (8p), and *ELAC2/HPC2* (17p11) (Figure 1) [12]. They are involved in the maintenance of genome stability, particularly in the processes of repair by homologous recombination (HRR), and repair of misfolded nucleotides (mismatch repair—MMR) and inter-chain DNA crosslinks. *BRCA1* and *BRCA2* genes predisposing to PC have also been found to be involved in pathways of control of checkpoints for DNA damage and replication fork protection during replication [13].

PC tumor cells are characterized by genome instability at both the chromosomal and gene levels. Genomic gains of chromosomes 7 and 8q and heterozygous losses of 8p, 13q, 16q, and 18 are often encountered in PC [14]. Additionally, PC tumors contain a large number of structural gene rearrangements, primarily translocations and deletions, the number of which increases markedly with disease progression. More than half of primary PC patients (up to 70%) show formation of the chimeric oncogene *TMPRSS2-ERG*—a product of fusion of the 5’-untranslated region of androgen-regulated gene *TMPRSS2* (chr21q22.2) with the ETS family of transcription factors genes, namely ERG (chr21q22.3), which results in aberrant expression of ERG [14,15,16,17]. *TMPRSS2:ERG* fusion is an early event in prostate carcinogenesis, it is absent in benign prostate hyperplasia (BPH) and in normal prostate tissues and is considered to be a diagnostic and prognostic marker for PC [18,19,20]. ERG-positive PCs are associated with a distinct spectrum of hypermethylation accompanied by epigenetically silenced genes, revealing a greater molecular and biological diversity [18]. Numerous studies have found a correlation between fusion of the *TMPRSS2* and ETS family genes (*ERG*, *ETV1*, *ETV4*, *ETV5*) and PC [18]. The fusion of these genes makes it possible for the ETS genes to be activated under by the action of activators of the *TMPRSS2* gene, thus triggering the cancer process in prostate cells [17]. *ERG* fusion genes are regulated by androgen receptors. In primary PC, *ERG* binds and involves the transcription factors AR, FOXA1, and HOXB13, leading to the activation of the Wnt and Notch pathways [21,22]. In *ERG*-positive PC cases with PTEN loss, PI3K signaling has been shown to be activated, resulting in increased proliferation and invasion. There are no specific prognostic biomarkers for *ERG*-positive tumors [23]. However, it is important to note that aggressive *ERG*-negative PC is associated with *SPOP*/*FOXA1* mutation, *CHD1* (5q15-q21) deletion, and *SPINK1* expression. The Gleason scale, PSA, and other biomarkers are exclusively for *ETS*-negative patients [24].

Primary PC germinal and somatic tumors are very genetically diverse. In recent years, due to the rapid development of molecular genetic methods, new approaches to genetic testing have been developed. Next-Generation Sequencing (NGS) has been introduced into clinical practice, represents a new approach to genetic testing. It makes it possible to perform panel testing of multiple genes for germline and somatic mutations, and to identify rare variants in genes with moderate penetrance [25]. The presence of these mutations leads to an increased risk of PC phenotype over a lifetime by 35–60%. The Cancer Genome Atlas (TCGA) consortium reported recurrent mutations or copy-number alterations within the following genes—*SPOP*, *FOXA1*, *IDH1*, *TP53*, *PTEN*, *PIK3CA*, *BRAF*, *CTNNB1*, *HRAS*, *MED12*, *ATM*, *CDKN1B*, *RB1, NKX3-1*, *AKT1*, *ZMYM3*, *KMT2C*, *KMT2D*, *ZNF770*, *CHD1*, *BRCA2*, *CDK12* [14]. Recent investigations have revealed that metastatic castration-resistant PC has a similar mutational landscape, but it harbors a higher mutational load and frequency of large-scale structural variants compared to primary PC [14]. However, the list of driver genes remained almost the same, including *PRAD*, *AR*, *TP53*, *MYC*, *ZMYM3*, *PTEN*, *PTPRD*, *ZFP36L2*, *ADAM15*, *MARCOD2*, *BRIP1*, *APC*, *KMT2C*, *CCAR2*, *NKX3-1*, *C8orf58*, and *RYBP*, albeit with AR gene mutations as a result of PC treatment [26]. Somatic alterations affecting AR expression, AR gene amplifications and treatment-associated mutations have been identified as being a major driver of castration resistance [26].

### 2.2. GWASs in PC

Genome-wide association studies (GWASs) with the use of multi-ancestry approaches are necessary to discover new risk variants for prostate cancer, to refine lead variants in known risk regions, and to develop genetic risk scores for PC for effective stratifying PC across populations. To date, GWASs and precise mapping studies of PC have mainly been performed in men from European, African, East Asian and Hispanic populations with the aim of identifying common genetic variants associated with disease risk across populations [27]. Approximately 270 loci harboring hundreds of single-nucleotide polymorphisms (SNPs) associated with PC risk have been revealed [27]. PC susceptibility loci were found on all chromosomes except chromosomes 15, 16, 21, and 23. PC risk-associated SNPs were highly enriched in noncoding cis-regulatory genomic regions [28]. Of 191 independent and replicated associations that reached genome-wide significance (i.e., *p* < 10^−8^), 123 (64%) were reported in populations of European origin, 21 (11%) in populations of Southeast Asia and 45 (24%) in other populations. Only one locus (17q21) reached genome-wide significance in populations of African ancestry [28]. Khan et al. showed little heterogeneity in associated PC variants across populations [3]. However, many loci found in populations of European or Asian ancestry were not replicated in populations of African ancestry, or the effect size was smaller (or directly opposite) for the race [3]. Several hypotheses have been proposed to explain the complexity of replication and differences in the magnitude of genetic effects depending on race and ethnicity. First, the underlying genetic predisposition, and hence the biology of PC, can differ fundamentally by racial or ethnic group [29]. This explanation has limitations, as it implies that the biological basis of PC depends on race or ethnicity. However, this hypothesis cannot be ruled out based on the available data. The most accurate explanation is that risk alleles and the underlying population structure of PC susceptibility loci differ by ethnicity or race, and that these differences likely affect the ability to detect genetic associations [29]. Polymorphic variants in the 8q24 region are associated not only with PC, but also with other cancers in different populations [29]. However, the biological mechanism of the influence of SNPs in this region, leading to prostate carcinogenesis, is unclear, since this region does not contain any coding DNA regions. The closest gene to this region is *MYC*, a proto-oncogene that is disrupted during carcinogenesis [29]. The 8q24 region may influence the expression of the *MYC* gene. The influence of 8q24 on Wnt signaling has also been determined [29]. As a result of the OncoArray project, 63 new loci associated with PC susceptibility have been identified, 52 of which were identified by imputation of OncoArray genotyping data [30]. A novel 6q27 variant (rs138004030) was found to be significantly associated with early PC onset [30]. Among the 63 new variants, several candidate genes were revealed; one of them is a missense variant of *ATM* gene rs1800057 [31]. Although this missense mutation was classified as “benign” in the ClinVar database, *ATM* has been implicated in the development of PC, and is strongly associated with the aggressive course of the disease. The ATM protein is a key checkpoint kinase that acts as a regulator of a wide range of downstream proteins, including TP53 and BRCA1, the CHEK2 checkpoint kinase, the RAD17 and RAD9 checkpoint proteins, and the NBS1 DNA repair protein [31]. Another missense variant (rs2066827) has been identified in the *CDKN1B* gene (cyclin-dependent kinase 1B inhibitor), which belongs to the Cip/Kip family of cyclin-dependent kinase inhibitors [29]. *CDKN1B* controls cell cycle progression to the G1 stage, and in vitro studies have shown *CDKN1B* levels to be associated with increased tumor size and tumor grade [29]. This particular variant has previously been associated with familial PC and disease progression. One more candidate gene, *RASSF3* rs7968403, has been identified. *RASSF3* is a plasma membrane GTP-binding protein and it is a member of the RAS signaling pathway that is aberrant in about one-third of cancers [29]. GWAS meta-analysis for Japanese and Chinese populations identified rs12791447/11p15.4 and rs58262369/14q23.2. The rs58262369 polymorphic locus is located in the 3’-untranslated region of *ESR2*, which encodes for estrogen receptor 2 [32]. Animal studies have shown high expression of *ESR2* in normal prostate epithelial cells, and *ESR2* knockout mice developed prostatic hyperplasia [32]. TCGA data showed that *ESR2* is upregulated in prostate tumors compared to normal prostate tissues; however, this does not correlate with increased mRNA expression [32]. rs12791447 is located in the intron of the *PPFIBP2* gene. According to TCGA, the expression level of this gene is significantly lower in tumors compared to normal prostate tissue [32]. To date, almost all available GWAS data have been obtained for the population of Western Europe and the white population of North America (79.8% [33]); therefore, further research is needed with the inclusion of populations from other regions, which will supplement existing information and will identify risk factors specific to different populations [34]. A population-specific approach and the ability of homogeneous populations to detect disease-specific SNPs is important for GWAS studies. At the same time, homogeneous population material provides a resource for checking previous GWAS results performed on mixed populations.

Future GWAS results will provide further support for a contribution of germline variation to ancestry differences in PC incidence. The clinical benefit of genetic risk scores profiling for targeted screening and early diagnosis also needs to be examined, and larger PC consortia in men of non-European ancestry will be required to identify additional risk variants, to improve precision of risk estimation and to enhance the predictive ability of the genetic risk scores across populations [33].

Despite the numerous genetic studies of PC, the results obtained during investigations are contradictory; they are poorly replicable in studies of different populations and are characterized by significant interethnic differences. The wide variability of clinical manifestations of PC indicates the need to study both general genetic risk factors for the development of the disease and specific factors predisposing to certain clinical and pathogenetic variants of disease course. It has to be taken under consideration that the immune system has a strong impact on the creation of permissive conditions for both the initial cell transformation and for the support of primary tumor growth and metastatic spread. Therefore, genetical aspects have to be considered with respect to the immunological landscape, which is affected by lifestyle, exposure to pathogens, and chronic low-grade inflammation driven by metabolic and stress factors. In this regard, the identification of risk factors specific to the development of PC in different populations is relevant. Identification of PC risk markers will make it possible to predict the risk of PC development with high accuracy and to develop preventive measures while taking into account the individual genetic characteristic of each patient.

It is necessary to confirm previously identified markers, and excluding false positive findings is an essential step in evaluating GWAS findings. Thus, replication and validation studies are an effective approach that will enable random findings and possible random associations to be disproved. Moreover, in order to bring the genetic aspects into the overall pathophysiological context, it is necessary to analyze how potential genetic predisposition affects PC risks in immunocompromised patients. However, such analysis is far from trivial, since a number of factors can be associated with compromised immune systems. For example, studies on PC in HIV-infected men are complicated by the significantly enhanced incidence of HIV infection in sexual minority groups, while important parameters such as sexual orientation are not routinely included in cancer registers [35]. An additional factor is that sexual minorities can experience poorer health-related quality-of-life outcomes compared to heterosexual men [35]. On the other hand, the incidence of PC might be decreased in men from sexual minorities living with HIV due to their shorter life expectations.

### 2.3. GWAS and PC Aggressiveness

The majority of men diagnosed with PC demonstrate indolent PC; therefore, detecting the genetic variants that are able to distinguish aggressive PC from non-aggressive one is of critical clinical importance for the prevention and treatment of the disease.

A few GWASs devoted to discovery of SNPs for prediction of aggressive PC have been conducted globally [36,37,38,39,40,41,42,43,44]. Risk assessment of PC is an important tool for distinguishing low-risk from high-risk PC, preventing PC overtreatment and helping to choose optimal and provide tailored treatment strategies for each patient. In 2007, Duggan et al. were the first to conduct a GWAS study for aggressive PC [36]. They performed a genome-wide association scan in 498 men with aggressive PC and 494 control subjects from Sweden. Among 60,000 SNPs they identified seven that had a similar (positive or negative) and statistically significant (*p* < 0.01) association with the risk of aggressive PC in both studies. Analyzing 1032 PC patients and 571 control subjects of European descent showed that only rs1571801 maintained a significant association after the validation stage and was associated with aggressive PC (one-sided *p* value = 0.004). (OR: 1.36; 95% CI: 1.13–1.63; *p* = 1.0 × 10^−3^). It is located in the *DAB2IP* gene, which encodes a novel Ras GTPase-activating protein and is a putative prostate tumor suppressor. Interestingly, the association with this SNP was stronger in patients with Gleason scores higher than 8 [36].

Sun et al., 2009 found an association between the rs9623117*C allele at 22q13 and the aggressiveness of PC [37]. The combined allelic test was highly significant, with *p* = 5.0 × 10(-7). The odds ratio (OR) of allele C for aggressive PC was estimated to be 1.18 [95% confidence interval (95% CI), 1.11–1.26]. The risk-associated variant was located within the genomic region of *TNRC6B*, a gene involved in miRNA-mediated mRNA degradation [37].

One of the studies enrolled 4829 and 12,205 patients with more and less aggressive disease, but did not collect data from control individuals without prostate cancer. They found that the frequency of the TT genotype of SNP rs4054823 at 17p12 was consistently higher among patients with more aggressive compared with less aggressive disease in each of the seven populations studied, (*p* = 2.1 × 10(-8)) under a recessive model, exceeding the conservative genome-wide significance level. The difference in frequency was greater between patients with high-grade, non-organ-confined disease compared with those with low-grade, organ-confined disease. This study demonstrated that inherited variants predisposing to aggressive but not indolent PC exist in the genome, and suggested that the clinical potential of such variants as potential early markers for risk of aggressive PC should be evaluated. The rs4054823 variant is located in an intergenic region, with the closest gene being *HS3ST3A1*, which encodes a heparan sulfate biosynthetic enzyme, a protein with no known relation to PC [38].

The investigation of FitzGerald et al., 2011 analyzed 387,384 autosomal single-nucleotide polymorphisms (SNPs), and rs6497287 located on 15q13 chromosome was confirmed to be most strongly associated with more aggressive (*p*(discovery) = 5.20 × 10(-5), *p*(validation) = 0.004) than less aggressive disease (*p* = 0.14) [39]. Additionally, rs3774315 on 3q26 was found to be associated with PC risk; however, the association was not stronger for disease that was more aggressive. Study cohorts included 202 PC cases with aggressive phenotype and 100 randomly sampled, age-matched prostate-specific-antigen-screened negative controls. Validation testing in an independent set of 527 cases with more aggressive and 595 cases with less aggressive prostate cancer, and 1167 age-matched controls confirmed the results obtained [39]. Another study enrolled patients with aggressive forms of PC and biopsy-proven normal controls ascertained from a PC screening program to GWAS [40]. They found significant associations between aggressive PC and five single-nucleotide polymorphisms (SNPs) in the 10q26 (rs10788165, rs10749408, and rs10788165, *p* value for association 1.3 × 10^−10^ to 3.2 × 10^−11^) and 15q21 (rs4775302 and rs1994198, *p* values for association 3.1 × 10^−8^ to 8.2 × 10^−9^) regions. Replication study proved combinations of these SNPs in 3439 patients undergoing prostate biopsy to be associated with aggressive forms of PC [40].

One of the most powerful investigations was performed by Amin Al Olama et al. [42]. They conducted bioinformatical analysis of previously published GWAS and enrolled datasets comprising 11,085 cases in which patients had an aggressive disease form and 11,463 healthy individuals. The analysis of 2.6 million SNPs revealed rs11672691 to be significantly associated with aggressive PC (OR = 1.12; 95% CI = 1.03–1.21). Validation studies comparing aggressive and indolent PC cases did not confirm the significance of the association between rs11672691 and aggressive PC risk [45]. The risk allele of rs11672691 (intergenic) was associated with an increased risk for PC-specific mortality and showed rs11672691 to be associated with both fatal and nonfatal PC [45]. This intergenic variant is located on the 19q13 chromosomal region within a long noncoding RNA gene—*PCT19.* According to several investigations, this SNP may be also involved in the regulation of *HOXA2*, *PCT19* and *CEACAM21* expression, in PC cell growth, invasion, metastasis and disease progression [46,47].

Gao et al. described a statistically significant association of rs11672691 with clinical features of aggressive PC, such as high tumor stage, high prostate-specific antigen (PSA) levels in 2738 men with disease progression, and the development of castration-resistant PC (CRPC) [47]. They analyzed the expression of two previously unknown PC genes, *PCT19* and *CEACAM21*, and found that aggressive CRPC is associated with the G allele rs11672691, which is involved in regulation of *PCT19* and *CEACAM21* gene expression and affects the cellular properties of the prostate tumor. It was hypothesized that analysis of the rs11672691 genotypes combined with analysis of *PCT19* and *CEACAM21* gene expression levels may predict PC recurrence and patient survival, which may be useful in determining further follow-up tactics [42,43,48,49].

The International Consortium for PC Genetics (ICPCG) performed GWAS of 2511 (unrelated) familial PC cases and 1382 unaffected controls, including 1394 aggressive PC cases and 1096 less-aggressive ones, but none of the SNPs (rs2735839, rs11672691, rs11704416, rs35148638, rs78943174) previously associated with aggressive disease were significant in this investigation [42,43,48,49].

Overall, all germinal mutations that have been highlighted by the studies in different geographical cohorts are found in genes related to the biology of cancer cells themselves, providing transformed cells with the enhanced potential to proliferate or migrate. However, such studies did not focus on the bioinformatics search in the screening data for the mutations in genes that make immune system more permissive for the transformed cells to start uncontrolled proliferation and invasion.

## 3. The Role of DNA Methylation in Prostate Carcinogenesis

DNA methylation has been known for over 2 decades and is considered to be one of the earliest and most stable changes in patients genome, occurring even before any signs and symptoms of prostate malignancy [49,50,51]. In normal somatic cells, most CpG islets are unmethylated. Aberrant methylation of the CpG islet of tumor growth suppressor genes can lead to the loss of its expression, enabling tumor initiation and progression. An understanding of the precise cause of aberrant methylation is necessary to identify the essential mechanism of cancer initiation and progression [52]. To date, a large number of methylated genes have been found to be associated with cancer. Tumor suppressor genes involved in DNA damage repair, cell cycle control, apoptosis, cell adhesion, and signal transduction are those most frequently hypermethylated in PC [53,54,55,56,57,58,59].

Taking into account the high frequency of aberrant DNA methylation in prostate cancer, such chromatin modifications are much more frequent compared to genetic alterations. DNA methylation patterns are attractive candidate biomarkers for the analysis not only of freshly isolated tumor samples, but also of archived FFPE samples. DNA methylation is catalyzed by a family of conserved DNA methyltransferases, and the main targets of methylation in DNA chains are CpG dinucleotides, where methyl groups are added to C5 cytosine residues in CpG dinucleotides [52]. The fact that DNA methylation patterns can be inherited makes this epigenetic modification stable in a series of cell divisions [52]. Several comprehensive reviews on aberrant DNA methylation in PC have been published recently, in which details about gene functions, their chromosomal localization and methylation patterns are comprehensively summarized and discussed [60,61]. Our review provides a short update on this topic.

DNA methylation analysis is a promising method that can be used to differentiate benign prostatic changes from prostate cancer. By examining the patterns of DNA methylation, we can identify distinct epigenetic signatures associated with prostate cancer, enabling differentiation between BPH and prostate cancer. Kim et al. analyzed tissue samples obtained during radical prostatectomy from a cohort of 53 patients, consisting of 42 cases of PC and 11 cases of benign prostatic hyperplasia [62]. They found that DNA methylation levels were able to accurately differentiate between PC and benign prostatic hyperplasia, and with high precision. The statistical significance of the prediction measured by the area under the curve (AUC) values for hypermethylated and hypomethylated DNA regions were 0.99 and 0.98, respectively, compared to the AUC of 0.79 for PSA. These findings suggested that DNA methylation analysis may provide a more accurate diagnostic tool for PC in comparison to the routinely used PSA test. This study also investigated the differences in DNA methylation patterns between early stage PC and its aggressive forms [62]. Early-stage PC exhibited global changes in DNA methylation, while aggressive or advanced forms of the disease were characterized by focal changes in methylation. The DNA methylation profiles of PC showed heterogeneity in terms of hypomethylation, whereas hypermethylation at CpG islands showed greater consistency among patients [62]. However, this study did not differentiate between the changes in DNA methylation in cancer cells and the cells of the tumor microenvironment. Identification of differences in DNA methylation between normal prostate samples and PC samples in all Gleason groups, as well as between PC samples and an extensive set of samples related to the urinary system, including normal prostate, bladder, and kidney samples, in 484 specimens revealed 263 prostate-cancer-specific DNA methylation biomarkers [63]. Correlation analysis between all pairs of sites and application of LASSO regularization ultimately identified six biomarkers with the potential to distinguish between (1) different Gleason stages, (2) PC and normal prostate tissue, and (3) PC and other cancers of urinary system [63].

Tolkach et al. found significantly higher levels of DNA methylation in the promoter region of the CD24 (sialoglycoprotein) gene, in tissue samples from PC patients (n = 30) compared to samples from patients with normal prostate (n = 35) and benign prostatic hyperplasia (n = 28) [64,65]. This reflects global changes in DNA methylation as the tumor progresses in prostate cancer. This study demonstrated increased CD24 promoter methylation and elevated CD24 gene expression correlated with poorer survival of PC patients [64]. For breast and ovarian cancer, it was shown that CD24 can be expressed in tumor cells and promote immune evasion by interaction with the inhibitory receptor sialic-acid-binding Ig-like lectin 10 (Siglec-10) expressed on TAMs [66]. It is of exceptional interest to find out whether a similar mechanism could be active in prostate cancer.

Methylation as a diagnostic tool can be utilized for rare morphological forms of prostate cancer. The analysis of DNA methylation patterns from 48 patients, including 9 with neuroendocrine PC and 39 with castration-resistant prostate adenocarcinoma, demonstrated 100% sensitivity and 90% specificity in detecting neuroendocrine carcinoma [67]. This finding has the potential to be used for the precise diagnosis of rare types of PC in order to make therapeutic decisions. Thus, several studies clearly indicate higher sensitivity and specificity of genome-wide methylation profiling as compared to existing diagnostic methods. Furthermore, such tests provide a minimally invasive procedure option by assessing methylation levels through liquid biopsy [68].

The DNA methylome can provide differential diagnostic value if analyzed at the systemic level, outside of tumor tissue. Analysis of the DNA methylome in the plasma of patients with localized (60 cases) and metastatic (175 cases) PC revealed that cell-free DNA methylome profiling is capable of detecting the difference between non-metastatic and metastatic patients beyond the analysis of tumor tissue [69]. Metastatic samples exhibited widespread hypermethylation accompanied by hypomethylation in pericentromeric regions. The analysis of the cell-free DNA methylome reflected clinical outcomes and was able to distinguish between non-metastatic and metastatic PC with a predictive accuracy of 0.989 [69]. Mehdi et al. investigated DNA methylation in peripheral blood T cells of men with positive PC biopsy compared to men with negative biopsy and benign prostate tissue as a control group [70]. The study revealed differential methylation at 449 CpG sites between control DNA and PCa T-cell DNA, showing a significant correlation with Gleason score (*p*  <  0.05) [70]. A total of 223 CpG sites were differentially methylated between controls and PC (*p* < 0.05), whereby the enrichment of pathways associated with immune surveillance was found, indicating a link between DNA methylation and development of PC [70].

In addition to peripheral blood samples, it is also possible to determine the levels of DNA methylation in urine samples. Investigation of the methylation status of the genes RARB, RASSF1, and GSTP1 in a cohort of 514 preoperative urine samples collected from treatment-naïve PC patients revealed DNA methylation changes in at least one gene in more than 80% of the urine samples obtained from patients diagnosed with PC [71].

In addition to its diagnostic value, DNA methylation analysis can be used as a prognostic biomarker. Biochemical recurrence is considered a decisive risk factor for clinical recurrence and metastasis of prostate cancer. A total of 262 distinct methylation sites associated with recurrence and non-recurrence were identified by Zhu et al., who analyzed DNA methylation data and corresponding clinical information from 480 PC samples obtained from the Cancer Genome Atlas [72]. Further application of LASSO Cox regression analysis selected an additional 35 key methylation sites as prognostic factors for biochemical recurrence-free survival [72]. A total of 1420 differentially methylated regions (DMRs) associated with prostate-cancer-specific mortality were identified by genome bisulfite sequencing and comparison of DNA methylation profiles in tissues obtained after radical prostatectomy between lethal and non-lethal patients, followed by validation in an expanded patient cohort [73]. Comparison of the study data with public PC datasets allowed the development of a prognostic panel consisting of 18 genes, and validation in an independent cohort showed a statistically significant association between hypermethylation at EPHB3, PARP6, TBX1, MARCH6, a regulatory element within CACNA2D4, and cancer-specific mortality [73].

In addition, there is a commercially available ConfirmMDx kit (MDxHealth, Irvine, CA, USA) to analyze DNA methylation to diagnose PC in histologically negative patients with suspected PC diagnosis based on ultrasound or MRI observations. The test includes the analysis of the methylation of three genes—GSTP1, APC or RASSF1—in biopsy tissue, is already used in clinical practice and has a sensitivity of 74.1% and a specificity of 60.0% [74,75]. Minimally invasive and noninvasive diagnostic systems based on the analysis of gene methylation in body fluid samples, such as blood, urine, semen, and others, are currently being actively developed. This appears promising for clinical use, considering their lower invasiveness and higher sensitivity and specificity compared to the established standard of PSA analysis.

## 4. The Role of Histone Modifications in Prostate Carcinogenesis

Another common type of epigenetic change in PC is histone code, which has been studied for the past 20 years [76,77]. However, the application of the considerable variability and flexibility of histone tail modifications has significant limitations compared to DNA methylation profiling. Posttranscriptional modifications of histones are essential for chromatin structure, determining its accessibility to the core transcriptional machinery and inducible transcription factors. The histone code serves as a crucial regulator in carcinogenesis [78,79]. Histone modification alterations can contribute to oncogenesis through enhanced regulation of transcription, replication, DNA repair, or cell cycle progression [78,80,81].

The most frequent modifications of histones are acetylation, methylation and ubiquitination of lysine, methylation of arginine, and phosphorylation of serine [82]. Histone methylation can occur on both lysines and arginines. Up to three methyl groups (me1, me2, me3) can be attached to each lysine residue. Alterations in the methylation status of lysine residues are related to transcriptional activity; for example, trimethylation of H3K27, H3K9 or H4K20 often results in gene inactivation, whereas trimethylation of H3K4, H3K36 or H3K79, in contrast, leads to an increase in transcriptional activity. Methylation of lysine at positions 4 (H3K4) and 27 (H3K27) of histone H3 is known to be a covalent modification associated with tumor development and progression [83].

Arginines, unlike lysines, can only be mono- (me1) and di- (me2) methylated. Two enzymes involved in arginine methylation are arginine methyltransferases 1 and 6, with methyltransferase 6 being a suppressor of transcription and arginine methyltransferase 1 being an activator of transcription [84]. Histone acetylation is associated with the addition of an acyl group (-COCH3) to lysine by acetyltransferases; deacetylation is associated with the removal of the acyl group by deacetylases. Lysine acetylation correlates with transcriptional activation [85].

There is increasing evidence for the involvement of histone modifications in the onset and progression of PC. Different types of modifications, especially methylation and acetylation, show different correlations between normal and cancer samples [86].

The investigation of histone H3K27me3 methylation profiles by chromatin immunoprecipitation reaction across the genome in 34 tissue samples (11 with Gleason score > 7, 10 with Gleason score ≤ 7, and 13 morphologically normal prostate samples) allowed the identification of an average of 386 genes marked with H3K27me3 in promoter regions in the healthy control group compared to 545 genes in the Gleason score ≤ 7 group and 748 genes in the Gleason score > 7 group. These results indicate that progressive diseases have a more extensive set of gene promoters enriched with H3K27me3 compared to normal tissues [87].

Deacetylation of H3K18Ac by deacytilase SIRT7 is crucial for maintaining key properties of PC cells, including exit from contact inhibition and anchorage-independent growth [88]. High levels of acetylated H3K27Ac are higher in primary cancer and metastases compared to benign tissues, and H3K27Ac tagging is essential for DNA demethylation [87].

Immunohistochemical evaluation of SIRT7 expression in malignant and adjacent normal tissues of the prostate gland in 57 patients with PC (sample included all pathological stages, mean age was 68 years, GS ≥ 6) revealed SIRT7 overexpression in tumors and a positive correlation with malignancy grade [89].

Decreased levels of acetylated histone H3K9ac have been associated with tumor progression, histologic grade, and clinical stage in prostate and ovarian tumors and a poor prognosis for these patients [90,91,92].

Immunohistochemical analysis of the tissue microarray, containing 23 benign tumor samples, samples of 48 patients with adenocarcinoma (Gleason < 7), and samples of patients with adenocarcinoma (Gleason ≥ 7), revealed significantly lower levels of repressing histone. This indicates dimethylation and trimethylation of H3K9 (H3K9me2 or H3K9me3) in prostate malignant tumor tissues [93]. H3K9me2 or H3K9me3 are epigenetic hallmarks of heterochromatin that can control lineage-specific gene repression in a tissue-specific manner [94]. Seligson et al. found a positive correlation of an activating histone maker H4R3me2 dimethylation with tumor malignancy level and predictive of tumor recurrence risk in 183 cases of primary PC (79 patients had Gleason score ≥ 7, 104 patients had Gleason score < 7), among which 171 cases had recurrent disease [90].

Despite the potential for studying histone modifications and their significant contribution to the fundamental understanding of PC carcinogenesis, investigating them remains challenging and costly, making the implementation of this knowledge in clinical practice currently less feasible.

## 5. MiRNA as a Prostate Cancer Biomarker

Despite the substantial progress made in the early detection and treatment of PC, it is still a challenging and relevant task to develop diagnostic panels based on new molecular markers that offer exceptional accuracy and specificity in discriminating aggressive forms of PC from indolent ones. Moreover, there is a clinical need to significantly enhance predictive value for tumor aggressiveness. The identification of new biomarkers exhibiting high precision and specificity is a condition for the timely detection of malignant transformations and for population-wide screening. Such biomarkers are crucial for disease prognosis and for therapeutic decision-making [95]. The current primary limitation in the diagnostic management of PC is the inadequate precision of biomarkers. Prostate biopsy, typically conducted when the serum prostate-specific antigen (PSA) level surpasses 2.5–4 ng/mL or abnormalities are identified during digital rectal examination, only confirms the presence of PC in a relatively low percentage of cases, ranging from 24% to 37% [96,97].

Epigenetic regulatory mechanisms play a pivotal role in the development of cancer, with epigenetic events often occurring at early stages of carcinogenesis [96,97,98,99]. The identification of miRNAs has emerged as a promising direction for advancing early diagnosis in PC, offering a potential breakthrough in improving the accuracy and effectiveness of early detection [100,101]. MiRNAs regulate multiple mRNAs, while each mRNA can be influenced by multiple miRNAs. Such network controls cellular processes that are crucial for tumor progression, like stemness, proliferation, differentiation, apoptosis, and metabolism. MiRNAs play a crucial role in fine tuning cellular physiology [101,102]. MiRNAs are responsible for regulation of around 60% of human genes [103,104,105]. The altered expression patterns of miRNAs lead to pathological conditions, including cancer [98].

MiRNAs are short, single-stranded RNA molecules that do not code for proteins. They exert their regulatory function by binding to the 3’-untranslated region (3’-UTR) of target mRNAs, thereby modulating post-transcriptional gene expression [98,99]. This binding can lead to translational suppression or degradation of the mRNA target. MiRNAs not only play a critical role in the biology of cancer cells, but can also control immunity [99]. Due to their involvement in prostate carcinogenesis, as well as their stability in biological fluids, miRNAs hold promise as non-invasive or minimally invasive diagnostic and prognostic biomarkers for PC [100,101].

### 5.1. MicroRNA Biogenesis

In mammals, a significant proportion of miRNAs are encoded by nucleotide sequences found within introns, accounting for over 90% of all miRNAs. In contrast, in fruit flies and invertebrates, this value is much lower, constituting only 14% of miRNAs [102]. The biogenesis of miRNAs can be divided in a series of distinct stages. The first step is the transcription of miRNA genes by RNA polymerase II, giving rise to precursor miRNAs (pre-miRNAs) with lengths typically spanning several hundred to several thousand nucleotides. In the subsequent stage, ribonuclease III (Drosha) and the RNA-binding protein Pasha (DGCR8) catalyze the cleavage of the miRNA precursor, resulting in the formation of primary miR (pre-miR) molecules. These pre-miR molecules typically have a length of approximately 70 nucleotides [103]. Next, exportin-5 transports pre-miR from the nucleus to the cytoplasm. In the cytoplasm, Dicer cleaves pre-miR, forming a mature RNA duplex of about 22 nucleotides. This duplex unwinds to generate the mature single-stranded miRNA, which regulates gene expression [104]. Mature miRNA, along with Ago2 and TRBP proteins, forms RISC (RNA-induced silencing complex). The passenger strand of the miRNA duplex is degraded, while the functional strand activates RISC and binds to target mRNA. This interaction leads to target mRNA degradation and translation inhibition, enabling miRNAs to regulate gene expression effectively [105]. Imperfect interaction between microRNAs and target mRNAs results in weak contact, leading to translation inhibition. On the other hand, a high degree of complementarity promotes strong binding and triggers proteolytic cleavage of the target mRNA with the involvement of the RISC complex. More recently, non-canonical biogenesis pathways for miRNAs, such as the synthesis pathway, have been discovered. However, the majority of pathways still rely on the involvement of Dicer enzymes for miRNA processing [106]. The mirtron pathway represents the first non-classical pathway identified for miRNA biogenesis. Unlike the classical pathway, it does not rely on the Drosha/Dgcr8 complex for the generation of pre-miRNA. However, it still involves the transport of XPO-5 and the cleavage of the Dicer enzyme, indicating its dependence on these factors for the processing of mirtron-derived miRNAs [107].

### 5.2. The Role of miRNA in Prostate Carcinogenesis

MiRNAs exhibit diverse profiles and functions that can be altered in various types of cancer [99]. MiRNAs can have a negative effect on gene expression by causing either a decrease or an increase in the affinity of miRNA sequences for their target mRNA. The impact of miRNA function is greatly influenced by the accessibility of the target mRNA. Consequently, individual miRNAs can have varying effects in different tissues, particularly in cancers arising from distinct cellular origins. The differential effects of miRNAs in various tissues and cancer types can be attributed to their unique regulatory networks and molecular landscapes [101,108]. Porkka et al. examined 319 miRNAs in PC using oligonucleotide array hybridization. Their study revealed differential expression of 51 miRNAs between benign and malignant tumors, with 37 downregulated and 14 upregulated miRNAs in carcinoma samples [109]. Not all of Porkka et al.’s findings have been consistently supported by subsequent studies. The field of miRNA expression profiling has expanded rapidly, with a growing number of platforms available for performing the analysis. Microarrays are commonly used for studying tumor-specific miRNAs, but they rely on the existing miRNA data, which may have limitations. This rapidly developing field now offers alternative approaches and technologies to better comprehend the complexity of miRNA expression patterns in cancer. Bioinformatic analysis identified a pool of miRNAs that influence PC carcinogenesis, only for the part of them the role in malignant prostate tumor development was experimentally demonstrated (Figure 2) [110]. Next-generation sequencing (NGS) technology revolutionized the identification of previously undiscovered miRNAs. In order to confirm miRNA expression profiles, real-time quantitative PCR (RT-qPCR) is considered the most suitable method for the accurate and precise quantification of miRNA expression [111].

### 5.3. Oncogenic miRNAs in Prostate Cancer

Malignant tumors of diverse origin are frequently characterized by decreased levels of miRNAs. This observation aligns with the state-of-the art opinion regarding the impact of miRNAs on cellular differentiation [112]. Certain miRNAs can exhibit oncogenic functions. Several such miRNAs have been identified, including the well-documented case of overexpression of oncogenic miR-181b in prostate cancer. He et al. conducted a study demonstrating that inhibiting miR-181b induced apoptosis in PC cells [113]. This highlights the potential therapeutic implications of modulating specific miRNAs in cancer treatment [113]. In their study, Leite et al. proposed that miR-21 exerts a negative regulation on RECK, a regulator of matrix metalloproteinases, consequently promoting the invasiveness of PC cells [114]. Furthermore, miR-21 has been implicated in promoting the aggressive potential of PC cells by regulating other tumor inhibitors, including the MARCKS protein (Myristoylated Alanine-Rich Protein Kinase C Substrate). MiR-21 targets the MARCKS protein, potentially leading to its downregulation and subsequent modulation of cellular processes associated with PC aggressiveness [115]. MiR-21 has also been found to negatively regulate other tumor suppressor genes, such as ANP32A and SMARCA4 [116]. By targeting these genes, miR-21 can inhibit their expression and potentially disrupt their tumor-suppressive functions [117]. Targeted inhibition of miR-21 has shown promising results in restoring apoptosis of cancer cells [115]. Blocking the activity of miR-21 leads to the re-establishment of the normal apoptotic pathway, leading to the suppression of tumor growth and progression [115]. Transfecting miR-21 into PC cells induces resistance to the anti-tumor drug docetaxel, reducing its effectiveness [117]. Multiple studies have demonstrated altered expression of miRNA-21 in PC, and a meta-analysis was recently performed evaluating its reliability as a diagnostic biomarker for the progression of PC [116]. The meta-analysis revealed miRNA-21 to be a reliable serum diagnostic biomarker candidate for metastatic progressive PC. Pooled sensitivity and specificity were 0.91 (95% CI 0.88–0.94, I2 = 0%) and 0.89 (95% CI 0.85–0.92, I2 = 44.8%), respectively. The positive and negative likelihood ratios were 7.18 (95% CI 4.31–11.96, I2 = 56%) and 0.11 (95% CI 0.07–0.16, I2 = 11.8%), respectively. The receiver operating characteristic (ROC) curve analysis yielded an AUC of approximately 97.4%. These findings supported the potential of miRNA-21 as a valuable diagnostic biomarker for assessing the progression of PC [116]. MiRNA-21 expression values have been found to correlate with the presence of castration-resistant prostate cancer (CRPC) and metastases, indicating its potential as a biomarker for assessing cancer progression. In a study by Seputra et al. in 2021, a total of 48 serum samples were examined for the expression of miRNA-21 by RT-PCR [118]. The cut-off value of miRNA-21 for distinguishing between benign prostatic hypertrophy and PC was in the range of 33.595–35.21. This cut-off value yielded a positive predictive value (PPV) of 87.5% and a negative predictive value (NPV) of 66.7%, with a *p*-value of 0.003. In the case of CRPC, the cut-off value for miRNA-21 was >35.21, with a PPV of 80% and an NPV of 58.3%, and a *p*-value of 0.04. The significant differences in miRNA-21 expression values between benign prostatic hypertrophy and CRPC, as well as between PC and CRPC, suggest the potential of miRNA-21 cut-off points as a diagnostic differentiator [118].

MiR-18a-5p has been identified as a tumor promoter in PC through various studies. The findings of Ibrahim et al. in 2021 support the notion that miR-18a-5p may act as a tumor-promoting factor in PC [119]. MiR-18a-5p expression increased in human PC tissues, while expression of SLC40A1 decreased. MiR-18a-5p promoted proliferation of PC cells in vitro, and miR-18a-5p downregulated SLC40A1 in PC cell lines. When SLC40A1 expression was restored, the effects of miR-18a-5p in PC cells was reversed. These findings indicate that miR-18a-5p contributes to the proliferation of PC cells, potentially through the downregulation of SLC40A1 [119]. Among Egyptian patients, miR-18a exhibited diagnostic significance in distinguishing PC patients from healthy individuals. MiR-18a displayed the highest AUC value of 0.966 (95% CI, 0.937–1.000), indicating its strong discriminatory power. On the other hand, miR-221 showed better differentiation between metastatic and localized PC, with a sensitivity of 92.9% at 100% specificity. When miR-18a and miR-221 were combined for patient differentiation, the sensitivity increased to 96.4% at a specificity of 100%, yielding an AUC of 0.997 (95% CI, 0.988–1.0) (*p* < 0.000) [119].

The miR-106b-25 cluster is located within intron 13 of the MCM7 gene, which encodes Minichromosome Maintenance Protein 7 [120]. Co-expression of MCM7 and miR-106b-25 affected development of prostatic intraepithelial neoplasia in transgenic mice. Additionally, miR-106b-25 negatively regulated PTEN expression [120]. MiR-106b-25 regulates expression of ZBTB4 at the post-transcriptional level. ZBTB4 functions as a tumor suppressor gene and inhibits expression of specific target genes through competitive binding with their promoter regions [120]. The miR-106b-25 cluster has been identified as a negative regulator of caspase-7 [121]. Regulation of miR-125b by androgen signaling remains a subject of debate. Some studies propose that the androgen receptor (AR) inhibits miR-125b expression, leading to the activation of specific mRNA transcripts [122]. In contradiction to this, Fredsoe et al. reported that androgen signaling actually stimulated miR-125b expression [123]. In this context, targeted inhibition of miR-125b resulted in reduced androgen-independent growth. These data indicate a potential positive regulatory role of androgen signaling on miR-125b expression [123]. Furthermore, miR-125b exerts a negative regulation on several proapoptotic genes, including p53, Puma, and Bak1. Through its action, miR-125b can inhibit the expression of these genes, crucial for promoting apoptosis [123]. MiR-4534 levels were found to be elevated in PC [124]. Interestingly, miR-4534 was hypermethylated in normal tissues compared to PC tissues/cells. The oncogenic effects of miR-4534 were attributed, at least in part, to its ability to downregulate the tumor suppressor gene PTEN. When miR-4534 was knocked down, it resulted in impaired cell proliferation, reduced migration/invasion capabilities, and induced cell cycle arrest at the G0/G1 phase, while also promoting apoptosis in PC cells [124]. Overexpression of miR-4534 was shown to induce pro-cancerous characteristics even in non-cancerous cell lines [124]. Furthermore, statistical analyses demonstrated that miR-4534 has the potential to independently differentiate between malignant and normal tissues, and positively correlates with poor overall survival and PSA recurrence-free survival [124]. Table 2 presents a comprehensive summary of oncogenic miRNAs.

### 5.4. Tumor Suppressor miRNAs in Prostate Cancer

The tumor suppressor role of miR-15a and miR-16 involves the regulation of oncogenes such as *BCL2*, *MCL1*, *CCND1*, and *WNT3A* [101,125]. The administration of a specific type of miRNA inhibitors called antagomirs, which are designed to silence miR-15a and miR-16, resulted in significant hyperplasia and increased disease severity in mouse PC models [101,125,130].Furthermore, the inhibition of miR-15a and miR-16 through the administration of antagomirs resulted in a decrease in survival, enhanced proliferation, invasion, and escalated severity of cancer disease in immunodeficient NOD-SCID mice [129]. Low expression levels of miR-15a and miR-16 have been observed in various malignancies, including chronic lymphocytic leukemia, pituitary adenoma, and prostate carcinoma. These microRNAs are encoded in the 13q14.3 region of chromosome 13 in humans, which is known to undergo deletions in several cancers, including PC, chronic lymphocytic leukemia, pituitary adenoma, and mantle cell lymphoma [131]. Zidan et al. found that miR-15a and miR-16-1 expression was significantly decreased in 80% of the examined PC samples compared to normal tissues [131].

Expression of miR-224, miR-16, miR-31, miR-125b, miR-143, miR-145, miR-149, miR-181b, miR-184, miR-205, miR-221, and miR-222 is known to be decreased in PC [125]. miR-143 and miR-145 have crucial functions in PC biology and function as tumor suppressors. They inhibit cell growth and promote apoptosis, contributing to the regulation of PC development [126]. miR-205 acts as a tumor suppressor in prostate PC, and its transfection into PC cells induces apoptosis. Notably, miR-205 is directly involved in stimulating the expression of tumor suppressor genes IL24 and IL32 [127]. MiR-205 also plays a role in downregulating Bcl-2, a protein involved in cell survival. When miR-205 is upregulated in PC cells, there is an increase in apoptosis, or programmed cell death, as can be seen with cisplatin and doxorubicin treatment [127]. MiR-574-3p, a microRNA that regulates Bcl-xL expression, is decreased in PC. Lower miR-574-3p levels result in the increased Bcl-xL, promoting cell survival and inhibiting apoptosis. Conversely, increasing miR-574-3p expression levels enhances apoptosis of PC cells [132].

The potential applications of exosomal miRNAs in the treatment and diagnosis of PC have generated significant interest. Numerous studies have highlighted the potential of extracellular vesicles (EVs) collected from blood as markers for diagnosing PC and determining the patient’s stage and prognosis. For instance, exosomal miRNAs miR-107 and miR-574-3p have been successfully quantified in the urine of men with prostate cancer. Recent findings have identified miR-375 and miR-141 as promising markers for high-risk PC [133,134]. Additionally, miR-141, miR-298, miR-346, and miR-375 have been found to be upregulated in serum samples from PC patients compared to controls [135,136]. MiR-141 has been further validated as a valuable diagnostic tool for PC patients [137]. Moreover, exosomal miR-141 has shown the ability to differentiate between localized PC and metastatic PC, with a higher AUC (0.869) compared to PSA (0.775). MiR-196a-5p and miR-501-3p also exhibited high diagnostic capabilities with AUC values of 0.73 and 0.69, respectively [134]. Li et al. identified potential non-invasive biomarkers in urine samples from PC patients, showing remarkable upregulation of exosomal miR-451a and exosomal miR-486-3p/5p compared to healthy controls [137]. Furthermore, exosomal miR-423-3p served as a predictive biomarker for early detection and castration-resistant PC [138]. The miR-125a-5p/miR-141-5p ratio was significantly higher in PC patients compared to healthy controls and exhibited better performance as an early PC biomarker than either miRNA alone [139]. Fredsoe et al. identified different miRNAs using qPCR in cell-free urine samples from patients with benign prostatic hyperplasia and clinically localized PC. They developed a new diagnostic model combining three miRNAs (miR-222-3p, miR-24-3p, miR-30c-5p) that distinguished benign prostatic hyperplasia from PC [123].

Yao et al. found that hsa-miR-182 was the most upregulated miRNA in PC tissue. Hirata et al. provided evidence that hsa-miR-182 promotes PC by targeting RECK, FOXF2, and MTSS1, which are tumor suppressor transcripts [140]. It has also been shown that miR-146a is upregulated in PC cell lines and tissues, indicating its potential as an onco-miR [141]. MiR-15b-5p and miR-106b-5p were significantly upregulated in aggressive PC tissue and correlated with disease aggressiveness [142]. Furthermore, a prognostic score combining the levels of miR-15b-5p and miR-106b-5p with serum PSA levels discriminated between indolent PC and an aggressive form with even higher accuracy [143]. A comparison of tissue and peripheral blood mononuclear cell samples revealed common downregulation of hsa-miR-494-3p, hsa-miR-3128, and hsa-miR-8084. Hsa-miR-494-3p targets three genes (HIF1A, NHS, INSL4), hsa-miR-3128 targets two genes (HIF1A, AVRP1A), and hsa-miR-8084 targets three genes (AVRP1A, NHS, INSL4). These findings suggest that hsa-miR-494-3p, hsa-miR-3128, hsa-miR-8084, and their target genes may play a crucial role in therapeutic and early diagnostic strategies for PC [144].

## 6. Pathogenesis and Staging

PC is the result of a complicated synergistic action of accumulated genetic changes aimed at boosting cell proliferation with respect to cell death. Early detection and identification of these occurrences is crucial for effectively managing the disease during its initial stages, facilitating the transition to an invasive tumor, predicting prognosis, and identifying optimal opportunities for therapeutic intervention. There is epidemiological evidence that indicates a potential association between symptomatic prostatitis and the risk of developing prostate cancer [145,146]. Data from 746,176 patients over 50 years of age diagnosed with PC from 2010 to 2013 were analyzed with follow-up until 2019 using the Korean National Health Insurance Service patient database [147]. The control groups were carefully matched based on age, presence of diseases such as diabetes and hypertension, and the Charlson comorbidity index. The incidence of PC was found to be significantly higher in the prostatitis group compared to the control group (1.8% vs. 0.6%, *p* < 0.001). The hazard ratio for developing PC was also significantly higher in patients with prostatitis (HR 2.99; 95% CI 2.89–3.09, *p* < 0.001). Furthermore, the HR for PC was notably higher in cases of acute prostatitis compared to chronic prostatitis (3.82; 95% CI 3.58–4.08; *p* < 0.001; HR 2.77; 95% CI 2.67–2.87, *p* < 0.001) [147].

The detailed mechanisms explaining the correlation between prostatitis and the risk of developing PC are not comprehensively understood at present. However, several factors have been proposed in which inflammation plays a prominent role. Prostatitis is characterized by inflammation of the prostate gland. Chronic inflammation can lead to DNA damage and alterations in cell signaling pathways [141]. Inflammatory processes in the prostate can result in the release of various cytokines and growth factors. These molecules can promote cell proliferation, angiogenesis (formation of new blood vessels), and tissue remodeling, which are all processes involved in tumor development and progression [148]. The immune response triggered by prostatitis may involve the recruitment of immune cells, such as macrophages and T-cells, into the prostate gland. Although these immune cells are intended to fight infection, their prolonged presence may contribute to chronic inflammation and create conditions favorable for evading immune surveillance [148]. It is important to note that while the association between prostatitis and PC risk was observed in epidemiological studies, not all individuals with prostatitis will develop PC. Additional research is warranted to gain a deeper understanding of the intricate mechanisms that underlie these associations and to identify potential therapeutic targets for intervention.

As stated earlier, uncontrolled activation of signaling pathways (which can be caused by inflammation), activated growth factors, in particular signaling through the lipid kinase group phosphoinositide 3-kinase (PI3K), contributes to carcinogenesis [149]. As part of the activation mechanism, the catalytic subunits of PI3K, directly bind to small GTPases, which appear to be members of the RAS subgroup. The role of PTEN in this pathway is to catalyze the opposite reaction by metabolizing PI3K [149]. Eventually, the triggering of signaling pathways leads to genomic instability, increased cell proliferation, cell invasion and migration (Figure 3).

In the majority of patients, localized PC is multifocal, due to the fact that cancer most often occurs in the peripheral zone of the prostate, the surrounding pseudocapsule is involved in 80% of clinically detected cases of cancer [150,151]. In the following stages, the cancer cells invade the seminal vesicles and paraprostatic tissue, regarded by the international TNM classification as T3, or the urinary bladder, levator muscles, external sphincter and/or anterior abdominal wall, which is the T4 stage [151]. The T1 stage of PC is typically assigned when the tumor is confined to the prostate gland and is clinically non-palpable and non-visualized. This means that the tumor will have been detected either during a prostate biopsy performed due to elevated PSA levels or incidentally during a transurethral resection. In other cases where the tumor does not extend beyond the prostate, it is classified as stage T2 [152]. Lymphatic metastasis involves the hypogastric, obturator, external iliac, presacral, common iliac, or retroperitoneal lymph nodes. Depending on the spread of the tumor to regional or distant lymph nodes, the diagnosis is N1 or N2, respectively. When PC spreads through the hematogenous route, it most often involves the bones of the axial skeleton, and less often the lungs, liver, and other soft tissues, which is assessed as stage M1 [153,154].

To evaluate the risk of PC both at the time of diagnosis and following treatment, various factors are taken into consideration, including the grading system, PSA level, tumor–node–metastasis classification (TNM), and the patient’s treatment history. These assessments are crucial for predicting the likelihood of adverse outcomes and guiding treatment decisions. In certain cases, such as intermediate-risk PC or high-risk PC, additional imaging studies may be recommended to further assess the extent and progression of the disease [146,152].

Malignant prostate tissue can exhibit indications of chronic inflammation. Histological analysis reveals the presence of inflammatory infiltrates, primarily consisting of CD3+ T lymphocytes, CD19 or CD20 B lymphocytes (10–15%), and macrophages (15%) [155]. The damage caused by the inflammatory response and subsequent chronic tissue healing is likely to contribute to the development of proliferative inflammatory atrophy (PIA).

Currently, there is inadequate evidence to be able to definitively establish a direct causal connection between prostate inflammation and prostate cancer. Ongoing investigations aim to explore a potential association between these two conditions; however, additional dependable and precise evidence is necessary to firmly establish this association. For instance, it has been suggested that the absence of glutathione S-transferase P1 (GSTP1) might be accountable for the progression from prostatic inflammation to high-grade intraepithelial neoplasia (HGPIN) and PC in individuals with a genetic predisposition [155]. Conducting further research will enable us to gain a better understanding of these mechanisms and their impact on the development of cancer.

Chronic inflammation preceding PC onset or cooperating with early stages of PC development can be an essential source for new biomarkers that can be used in combination with PCs or other relevant cancer cell-derived factors. Such biomarkers can be released by immune cells in TME, or can be identified in circulating immune cells, in particular in monocytes. We have previously demonstrated that monocytes provide clinically valuable information for patients with breast and colorectal cancer [156,157,158]. In breast cancer CD163+ CD14^low^CD16+ and CD163+ CD14+CD16+ monocytes were indicative for the presence of the malignancy, and CD14^low^CD16+HLA-DR + monocytes were predictive for the good response to neoadjuvant chemotherapy [156]. In colorectal cancer (CRC), the monocyte biomarkers showed distinct correlations with metastatic processes [157]. Elevated levels of CCR2+ monocytes in rectal cancer were associated with the absence of lymphatic and hematogenous metastasis. Conversely, in patients with colon cancer, CD163+ monocytes showed a positive correlation with lymphatic involvement [157]. In this study, we also performed full transcriptome profiling in circulating monocytes by NGS, and identified PFKFB3, activator of glycolysis that is currently an attractive candidate for several solid cancers [157]. PFKFB3+ monocyte-derived macrophages massively infiltrated tumor in the colon, and Nanostring spatial profiling identified a correlation of PFKFB3 with tumor-promoting properties of TAMs in colon but not in rectal cancer. Monocyte-expressed PFKFB3 was indicative for tumor relapse specifically in colon but not rectal cancer [157]. Monocytes are not only indicators of systemic immune status, but can be also be programmed by systemic changes caused by all types of anti-cancer therapy: surgery, radiotherapy, chemotherapy and immunotherapy [158]. In cardio-metabolic disorders, monocytes are crucial biomarkers predicting progression of vascular complications related to the pathological lipid metabolism, and key role of monocyte-derived foamy macrophages is established [159,160,161]. Considering foamy macrophages in PC recently identified by us and others, the potential of circulating monocytes as predictors of therapy efficiency and biomarkers that can be used for the personification of therapeutic schemas is only emerging, but has a solid theoretical background. For PC, the clinical value of monocyte programming and subpopulations urgently needs to be identified.

## 7. Methods for Diagnosing Prostate Cancer

### 7.1. Currently Used Diagnostic Approaches

Clinical guidelines from various countries and leading cancer societies recommend similar algorithms for the diagnosis of PC [146,152]:Digital rectal examination (DRE); and/orTransrectal ultrasound (TRUS).Serum PSA (prostate-specific antigen): total PSA and the ratio of free PSA to total PSA.Biopsy confirmation.

Table 3 summarizes the information about the sensitivity (the ability of the test to correctly identify patients with the disease) and specificity (the test’s ability to correctly identify patients without the disease) of the PC tests available for clinical application.

Tumor biopsy testing for somatic gene mutations (e.g., *BRCA1*, *BRCA2*, *ATM*, *PALB2*, *FANCA*, *RAD51D*, and *CHEK2*), MSI, or dMMR should be considered as additional diagnostic options for patients with regional or distant metastases [146]. In Table 3, data on the sensitivity and specificity of each type of diagnostic method are summarized, along with their advantages and disadvantages. Each type of test has limitations and can miss a case of prostate cancer; only the combination of several tests can provide the most complete picture. The maximal sensitivity of a single does not exceed 77%, and the maximal specificity of a single test is below 50%. Therefore, it is necessary to improve the test systems or their combination to allow precise clinical decision.

### 7.2. Necessity of Early Screening

Early PC can be asymptomatic, so screening is necessary. The gold standard for screening now is PSA testing and Digital Rectal Examination [152,162]. Since the introduction of PSA testing and subsequent biopsies, the incidence of PC in the USA has doubled, beginning in the late 1980s [163,164]. A UK interview-based study used the EPIC-26 questionnaire to analyze 3523 patients 18–42 months after they were diagnosed with prostate cancer. This study revealed that quality of life was higher in patients diagnosed with PSA-screened PC compared to patients diagnosed due to symptoms of PC [148]. PC was detected through PSA testing in 31.3% of cases, while patients presented with symptoms in 59.7% of cases. In a multivariate analysis, men with symptoms reported more severe problems with urinary incontinence, bladder and bowel function, sexual function, and vital/hormonal function compared to men whose PC was found via the PSA test. These differences were consistent among respondents of different ages, stages, Gleason scales, and treatment types [148].

Early detection of PC improves overall patient survival in low-risk patients, but has no effect on the survival of patients with metastatic cancer [149,150,165]. Since the management strategy for low-risk patients is active monitoring, careful patient selection and accuracy in risk assessment require new, more precise and more specific biomarkers, since randomized controlled trials have shown that PSA screening has no effect on overall mortality or prostate cancer-caused mortality [166]. According to a meta-analysis of five randomized controlled trials that encompassed a total of 721,718 men, screening is likely to result in a minor decrease in specific mortality over a 10-year period. However, the analysis did not identify any impact on overall mortality [151]. The meta-analysis revealed that screening for PC does not have any impact on all-cause mortality and may not significantly affect prostate-specific mortality or only result in a slight reduction in prostate-specific mortality. Utilizing Der Simonian and Laird’s inverse of variance random effects modeling, it was estimated that of every 1000 men screened, approximately 0.1% may experience hospitalization due to biopsy complications, 0.3% may encounter a decline in quality of life due to bladder control problems, and 2.5% may experience erectile dysfunction [151]. In addition to the side effects caused by screening procedures, it is necessary to consider the patient’s psychological discomfort due to false positive PSA tests.

Moreover, there is substantial evidence supporting the advantages of measuring baseline PSA levels in middle-aged men as a means to assess their future risk of developing PC [149]. Examining 13 years of data from a cohort of 10,968 men aged 55 to 60, who were part of the PC screening group, revealed a correlation between baseline PSA levels and the likelihood of developing prostate cancer. Notably, men with baseline PSA levels below 2.00 ng/mL exhibited a significantly lower risk of PC [154].

From the standpoint of reducing mortality, overtreatment, which refers to unnecessary or excessive medical interventions, offers little or no benefit. Several years following radical prostatectomy or radiation therapy for high-risk prostate cancer, a considerable number of men continued to experience substantial deterioration in various functional domains, including sexual, urinary, and bowel function [7,150,165,166]. Modeling of the lifelong consequences of annual PSA screening at the age of 55–69 years compared with the absence of screening estimated the loss of 23% of the years of life obtained during screening, mainly due to deterioration in the quality of life due to long-term side effects from treatment [149].

The study covered a cohort of 182,160 men aged 55 to 69 from eight European countries, and calculated the number of men that should be called for screening to avert one death from PC [167]. They used the Wald method to calculate confidence intervals for the difference in the risk of death. The results of this calculation demonstrated that the prevention of one death from PC requires the screening of PSA levels for 9 years in 1947 men, for 13 years in 742 men, and for 16 years in 570 men [167].

The search for immunological biomarkers in PC has been predominantly focused on regulators of inflammation, mainly cytokines and tumor-infiltrating lymphocytes. Several pro-inflammatory cytokines have emerged as potential biomarkers, as inflammation has been implicated in tumor initiation and progression [168,169]. For instance, increased levels of IL-8, TNF-α, and MCP-1 (CCL2) have been associated with worse overall survival in metastatic PC patients undergoing androgen deprivation therapy [170]. Transforming growth factor-β1 (TGF-β1), involved in cell-mediated immunity, has been linked to increased Gleason score, biochemical recurrence after radical prostatectomy, and disease progression [171,172]. Cytokines like IL-8 and stromal-cell-derived factor-1 (SDF-1) have also been directly linked to PC progression [173]. Toll-like receptors (TLRs), responsible for recognition of invading pathogens and metabolic ligands, have been a subject of research, with increased TLR-9 observed in poorly differentiated prostate neoplasms [174]. Dysregulation of TLRs, either up- or downregulation, has been associated with a high rate of PC recurrence [175]. Cells of the tumor microenvironment have also gained attention as biomarkers in PC. Prostate cancer tumors are often infiltrated by regulatory T cells, PD-1+ cells, and PD-L1+ cells. Elevated PD-1 expression in T cells has been linked to shorter biochemical survival and increased CD8+ lymphocyte density [176,177]. In a study exploring genetic and immunological approaches, PLK1 emerged as a potential contributor to PC progression based on the LnCeVar database [178]. A study utilizing double immunofluorescence tagging of CD163, an M2 macrophage marker, and PLK1 was conducted on human PC tumor tissue microarrays. The results demonstrated a positive association between PLK1 and CD163 in PC samples, while no such correlation was observed in benign hyperplasia samples [178].

### 7.3. Biopsy Examination and How TAMs Can Help

Prostate biopsy is a medical procedure used to diagnose prostate cancer; it involves the removal of a small tissue sample from the prostate gland, which is then examined under a microscope for the presence of cancer cells [179]. To assess the degree of differentiation prognostically, pathologists evaluate PC biopsies. When examining biopsy samples, an assessment is made of the shape, size of cells, nuclear changes, and the presence of anomalies in cellular structure [179]. Additionally, the degree of differentiation of cancer cells is evaluated. Furthermore, the quantity and distribution of cancer cells in the sample, as well as the presence of invasion into surrounding tissues, are assessed. This helps to determine the stage of PC and its potential for spread or metastasis [179].

The histological evaluation of PC samples also includes a description of specific pathological features, such as the presence of glandular structures, cribriform patterns, perineural invasion, and others. Donald Gleason developed a scale to standardize the evaluation of architectural details in malignant glands at low and medium magnification levels, aiming to unify the examination of samples [180]. The Gleason scale is utilized to classify cancerous tissue based on microscopic evaluation of histopathological features, ranging from poorly differentiated (the highest grade) to highly differentiated (the lowest grade). In 2014, the International Society of Urological Pathology (ISUP) reorganized the classification system into groups 1–5 according to their updated classification [181]. The Gleason classification of acinar adenocarcinoma of the prostate is one of the earliest and most successful applications of evidence-based medicine in routine clinical practice. The original Gleason system has shown excellent correlation with clinical outcomes [182,183].

In recent years, there have been notable advancements in the field of PC diagnosis and treatment. These advancements include the emergence of various diagnostic tests utilizing novel biomarkers. The primary objective of these tests is to enhance the specificity and sensitivity of PSA screening, ultimately leading to improved accuracy in PC detection. Additionally, these advancements aim to optimize treatment strategies, ensuring enhanced efficiency and effectiveness in managing the disease. In light of these advancements, it is becoming increasingly crucial to consider the incorporation of newly identified biomarkers to further enhance diagnostic accuracy and optimize patient management.

Emerging biomarkers like PHI, TMPRSS2-ERG fusion gene, 4K test, and PC3 test enhance PC diagnosis, alongside PSA testing, improving accuracy and aiding early detection for timely treatment decisions [184]. The 4K test, for instance, utilizes a panel comprising total PSA (tPSA), free PSA (fPSA), intact PSA (iPSA), and human kallikrein 2 (hK2). By analyzing these biomarkers, it becomes possible to differentiate between various causes of elevated PSA values, offering valuable insights into the presence and characteristics of PC [185]. Unbound PSA is known as free PSA (fPSA), which is about 5–35% of total PSA. The investigation of different forms of PSA, such as free PSA (fPSA) and intact PSA (iPSA), has provided valuable information for the identification and stratification of prostate cancer. While fPSA represents the unbound fraction of PSA, accounting for approximately 5% to 35% of total PSA, iPSA demonstrates a higher ratio in men with cancer compared to those without tumors [174]. Furthermore, human kallikrein 2 (hK2), a closely related protein to PSA, exhibits structural and functional similarities, offering additional insights into PC detection [186]. These biomarkers can help to identify high-grade PC in men who have not previously been tested for elevated PSA. The PHI (prostate health index) defined by the formula [([-2]proPSA/free PSA) × √total PSA] was developed for prognostic purposes [149,187]. The PHI, compared to other PSA-related tests, has been shown to better differentiate PC from benign prostatic hyperplasia, and thus to prevent unnecessary prostate biopsy [85,184]. The PHI test makes it possible to detect the progression of PC with active monitoring of the condition. Additional non-invasive biomarkers used for diagnostic purposes are *TMPRSS2-ERG* Fusion and PC3 Progensa PC Antigen 3 based on urine [184]. One of the markers of castrate resistance in PC is the level of *TMPRSS2-ERG*. Serine protease 2 (TMPRSS2), as well as the *ERG* gene, can be found in 50% of patients with prostate cancer, and overexpression of the PC3 or *DD3* gene (specific non-coding mRNA) is observed in more than 95% of cases of primary PC [188]. However, the assessment of the above markers in comparison with the prostate specific antigen has a lower sensitivity. The accuracy of PC biomarkers is critical in determining the further tactics, and the transition to invasive methods of examination [188].

State-of-the art gene expression-based tests like Oncotype DX Prostate and Prolaris aid in treatment decision making for prostate biopsies during active surveillance. Decipher is designed for high-risk surgical specimens. Decipher uses 22 genomic classifiers (GCs) with 19 genes to assess genomic risk on a scale of 0 to 1. It analyzes the entire transcriptome and RNA biomarkers associated with cell proliferation, differentiation, motility, and immune response [189]. Decipher predicts relapse or metastasis after radical prostatectomy in patients with adverse comorbidities [190]. GC-based systems outperform conventional parameters in metastasis prediction [191]. Oncotype DX Prostate predicts adverse events after radical prostatectomy in low-risk patients with 10–20-year life expectancy. It uses 12 genes associated with morphological changes, proliferation, and androgen signaling [192]. Prolaris calculates a risk score based on 31 cell cycle progression genes. It benefits post-biopsy prostatectomy patients and untreated low-risk patients [193]. CCP score predicts risk and mortality in patients treated conservatively [193].

As can be seen from the above, the accuracy of PC biomarkers is critical in determining the further tactics to be employed, and the transition to invasive methods of examination. To decide the necessity of screening, it is necessary to take into account and individually discuss the strategy with transgender patients taking or not taking hormone therapy, since the risk of PC for such patients has not been determined [88,89]. However, the biopsy examination in current clinical practice does not consider the evaluation of any immunological parameters in the tumor microenvironment. Tumor-associated macrophages massively infiltrate solid tumors, and in several cancers, including prostate cancer, poor prognosis is correlated with increased numbers of TAMs, identified in the majority of studies by the highly specific macrophage marker CD68, which belongs to the scavenger receptor family [194].

However, there are also cancer types, or specific intertumoral compartments, where levels of CD68+ TAMs in parenchyma in breast cancer have negative correlations with lymphatic metastasis, and in colorectal cancer, total CD68 levels positively correlate with overall survival and good prognosis [195,196,197,198]. Therefore, a number of studies have been performed focusing on specific TAM biomarkers, including scavenger/endocytic receptors, such as CD206, stabilin-1, MARCO, and others, which have demonstrated specific correlations with stages, metastasis and even therapy responses for lung, breast, colorectal, ovarian, prostate and other types of cancer [199]. However, scavenger receptors (SRs) are limited in the demonstration of specific TAM functional polarization to the high redundancy of SRs’ ligand repertoire and functions [195]. More specific biomarkers with clear distinct functions are needed to distinguish between tumor-promoting and anti-tumor TAMs. Here, Chitinase-like proteins, potent regulators of angiogenesis and immune cell recruitment, can offer much more precise biomarker value [196]. Metabolic regulators, as identified for colorectal cancer, also show great potential [199]. In the case of prostate cancer, we have observed, even on the morphological level, two major phenotypes of TAMs: regular in size and shape, and large foamy-type TAMs [198]. Identification of the molecular profile and functions of these two major phenotypes of TAMs is needed to decipher their biomarker potential.

TAMs play a significant role in PC pathology. Increased TAM density in PC demonstrated a correlation with higher Gleason score and shorter cancer-specific survival. Additionally, TAMs have been associated with the activation of osteoclast-related pathways, which is particularly relevant considering the propensity of PC to metastasize to the bones [199]. Understanding the involvement of TAMs in PC holds clinical importance for developing targeted therapeutic approaches.

The limited integration of these new biomarkers in routine clinical practice can be attributed to several factors. Further validation and standardization across different populations and settings, cost-effectiveness, and accessibility challenges, as well as interpretational complexities, hinder their widespread use. The reliance on PSA as the primary screening tool and the need for a shift in established practices also contribute to their limited utilization. Continued research, standardization efforts, and evidence-based guidelines are crucial for facilitating their integration into PC care.

**Table 3 ijms-24-12797-t003:** Diagnostic procedures for prostate cancer according to the international clinical guidelines [158,159].

Diagnostic Method	Principle	Sensitivity (0–1)	Specificity (0–1)	False-Negative Cases (%)	False Positive Cases (%)	Benefits	Limitations	Reference
Digital Rectal Examination (DRE)	Palpation of the lower part of the rectum, pelvis and lower abdomen	0.51	0.59	-	-	availability and affordability non-invasive	low sensitivity lack specificity more than 60% are identified as asymptomatic	[166,200,201]
Prostate Specific Antigen (PSA)	Venous blood sampling for prostate-specific antigen, a glycoprotein expressed in both cancerous and normal columnar prostate epithelial cells.	0.21–0.5	0.91	10–15%	-	availability and affordability	lack specificity predictive accuracy of 8% to 10%	[166,202,203]
Transrectal Ultrasound Scan (TRUS)	Ultrasound examination of the prostate with insertion of the sensor into the rectum.	-	-	11.34–29.31	4.61–6.11%	availability and affordability non-invasive	lack specificity	[166]
Transrectal biopsy (TRB)	Tissue sampling with a thin needle that is inserted through the rectum into the prostate.	0.53	1	11–46%	-	Availability affordability	most lesion are small and sometime located in regions that are not identifiable complications of prostate biopsy (e.g., infection, pain, bleeding, urinary obstruction)	[141,147,166,200,204,205]
MRI-guided biopsy	MRI-guided sampling of prostate tumor tissue	0.77	1	6%	4.2%	accuracy	complications of prostate biopsy (e.g., infection, pain, bleeding, urinary obstruction); expensive	[141,147,204]
MRI	Creation of detailed volumetric images of areas using a magnetic tomography	0.67	0.92	2.7%	44.1%	non-invasive	expensive	[141,147]

“-” no data available.

### 7.4. State-of-the Art Imaging for Staging and Metastasis Detection

A multicenter confirmatory study was conducted to assess the diagnostic accuracy of MP-MRI and TRU-biopsy in a paired-cohort setting [206]. The study included a total of 740 men with prostate-specific antigen concentrations up to 15 ng/mL, who underwent MRI followed by TRU biopsy. The performance and reporting of each test were conducted blindly to the results of the other tests. Clinically significant cancer was defined as a Gleason score of 4 + 3 or higher. MP-MRI demonstrated higher sensitivity (93%, 95% CI 88–96%) for detecting clinically significant cancer compared to TRU biopsy (48%, 42–55%; *p* < 0.0001), but lower specificity (41%, 36–46% for MP-MRI versus 96%, 94–98% for TRUS biopsy; *p* < 0.0001). Among the 740 patients, 44 (5.9%) reported serious adverse events, including 8 cases of sepsis [206].

Advanced imaging techniques such as CT, MRI, and PET have significantly improved the ability to detect metastases and accurately stage PC. In particular, MRI has shown high accuracy in reducing the need for unnecessary biopsies compared to the standard transrectal ultrasound (TRUS) approach. To standardize the interpretation of MRI findings, the Prostate Imaging Reporting and Data System (PI-RADS) was introduced by the European Society of Urogenital Radiology. PI-RADS provides a standardized framework for assessing and reporting MRI results, enhancing consistency and facilitating effective communication among healthcare professionals. Key elements of assessment include prostate volume measurement, lesion mapping, lesion measurement, and lesion assessment [207].

However, the high cost and low availability of these methods is an obstacle to accurate diagnosis (Table 2).

TAMs play a significant role in PC therapy. TAMs are key components of the tumor microenvironment and have both anti-tumor and anti-inflammatory effects. In prostate cancer, TAMs interact with tumor cells and immune cells, such as T lymphocytes, influencing disease progression [208]. One important function of TAMs in PC is regulation of inflammation and tissue remodeling. TAMs express immunosuppressive markers, such as PD-L1, which suppress the activation of effector T cells and facilitate immune evasion by the tumor. TAMs can undergo phenotypic changes during cancer progression, transitioning from an anti-tumor state to an anti-inflammatory state, thereby enhancing immunosuppressive effects and promoting tumor growth [208].

Modulating TAM activation may improve the effectiveness of immunotherapy in prostate cancer. Targeting signaling pathways or using specific drugs to modulate TAM activation could enhance immunotherapy efficacy and improve treatment outcomes. However, a deeper understanding of TAM interactions with tumor cells and the tumor microenvironment is needed to develop more effective immunotherapeutic strategies in prostate cancer.

### 7.5. Raman Spectroscopy

Raman spectroscopic analysis is an advanced optical diagnostic method that is utilized in the examination of tissues and biofluids for in vitro and in vivo diagnostics. This technique provides real-time results and offers a molecular portrait of tissue in malignant diseases, including prostate cancer. The basic principle of Raman spectroscopy is associated with the interaction of light with the molecules of substances contained in the analyzed samples [209]. Raman spectroscopy has been positively evaluated as a diagnostic method for differential analysis of malignant and non-malignant tumors of the prostate tissue. Aubertin et al. conducted a study of 32 fresh prostatectomy tissue samples with Raman spectroscopy to detect and analyze the severity of PC [210]: from samples of fresh prostate tissue, sections were obtained, from each of which 20 to 50 scattering spectra were obtained, the total number of which was 947 spectra. Using Raman spectroscopy, it was found that the sensitivity in identifying the prostate among non-prostatic tissue was 82%, the specificity was 83%. During the analysis, notable findings included the ability of the method to differentiate benign prostate tissue from malignant tissue with a sensitivity of 87% and a specificity of 86% [211]. Pinto et al. integrated Raman spectroscopy into the surgical workflow of robotic radical prostatectomy, incorporating both in vivo (n = 4) and ex vivo (based on 599 spectra from 20 prostatectomy specimens) analyses. The aim was to differentiate between cancerous and benign prostate tissue using this technique [211]. In a study conducted by Medipalli et al., the Raman spectra of plasma samples from 43 patients diagnosed with PC and 33 healthy volunteers were compared. The analysis revealed an increase in the bands associated with nucleic acids in the Raman spectra of plasma from patients with PC [212]. In interpreting the results, it is worth noting that the increase in nucleic acid concentration may be associated with abnormally increased gene expression, which correlates with increased release of nucleic acids due to cell death. In patients diagnosed with prostate cancer, compared with healthy people, absorption bands corresponding to lipids are more common. This may be due to the fact that the risks of tumor progression and treatment resistance are associated with high levels of lipids and cholesterol in the tumor.

Raman spectroscopy is a minimally invasive analysis technique that can be used to analyze samples of urine, blood, saliva, and other body fluids. Del Mistro et al. conducted an experimental surface-enhanced Raman scattering (SERS) analysis of urine samples from nine patients diagnosed with PC and nine healthy volunteers. Using machine learning algorithms, the obtained spectra were divided into two groups (PC or healthy people), resulting in a sensitivity of the analysis of 100%, a specificity of 89% [213]. Similarly, Ma et al. used SERS to obtain urine spectra of 75 patients, of which 12 patients with recurrent PC and 63 patients with non-recurrent PC [214]. While comparing Raman bands in patients without recurrence and with recurrent prostate cancer, it was found that the latter have enhanced Raman bands associated with lipids, proteins, amino acids and DNA [214]. The review conducted by Chen et al. demonstrates the most complete information about Raman scattering methods based on urine [215]. The measurement of prostate-specific antigen (PSA) protein in serum is widely used as a common method to identify men at risk of developing prostate cancer. However, the PSA test has certain limitations, particularly in terms of type I errors [202]. Falsely elevated PSA can be observed with prostatitis, benign prostatic hyperplasia, and urinary tract infections [203]. Chen et al. conducted a study to explore the potential of serum surface-enhanced Raman scattering (SERS) as a screening tool for differentiating between PC and BPH in patients with PSA levels in the 4–10 ng/mL range, which poses a diagnostic challenge [216].

Raman imaging techniques allow specific identification of individually pure substances, often without the need for contrast agents. This presents great promise for in vivo applications in disease diagnosis, treatment outcome monitoring, disease prognosis, and disease staging. The development of specialized nanoparticles in the surface-enhanced Raman scattering (SERS) process enables high sensitivity and specificity for diagnostic applications in diverse biological fluids.

## 8. Conclusions

In conclusion, the current limitations of clinical and pathological parameters in accurately differentiating PC types present a significant obstacle in preventing unnecessary treatments. The application of modern molecular genetic techniques has provided an opportunity to gather comprehensive data encompassing genomic, transcriptomic, epigenomic, proteomic, and metabolomic profiles from various sources such as biopsies, prostatectomies, and individual cells. The integration of these data into clinical practice is of utmost importance [7]. However, each tumor develops in close contact with the immune system, both systemic and local, where the immune system can be (1) pre-programmed to be more cancer permissive or restrictive; (2) programmed by growing tumor; (3) programmed by cancer therapy. Such programming can be on epigenetic, transcriptional and metabolic levels, and each level can offer clinically applicable options [109,150]. Thus, the immune biomarkers, in particular biomarkers provided by innate immunity (circulating monocytes and TAMs), are essential for significantly enhancing the precision of the diagnostics and achieving maximal therapeutic efficacy. Advanced and precise technologies of spatial transcriptomics and spatial epigenomics, single cell analysis of transcriptome, epigenome and metabolome open the way to identifying crucial detrimental immune cell subpopulations as well as key immune biomarkers that can be translated to clinics. However, the application of such technologies requires, in most of cases, prospective patient sample collections and substantial funding.

By harnessing this wealth of information, we can gain a better understanding of the disease’s variability and tumor progression, thereby identifying valuable biomarkers for effectively managing newly diagnosed cases and tailoring treatment strategies to individual patients. This approach holds great promise in optimizing PC care and ensuring that patients receive the most appropriate and personalized treatments.

## Figures and Tables

**Figure 1 ijms-24-12797-f001:**
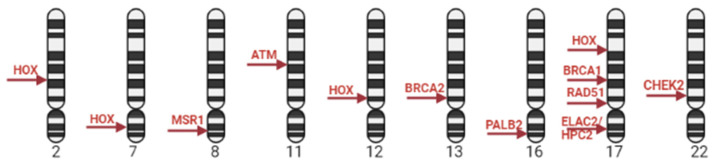
The location of mutations in DNA repair genes detected by sequencing DNA samples from men with a family history of prostate cancer.

**Figure 2 ijms-24-12797-f002:**
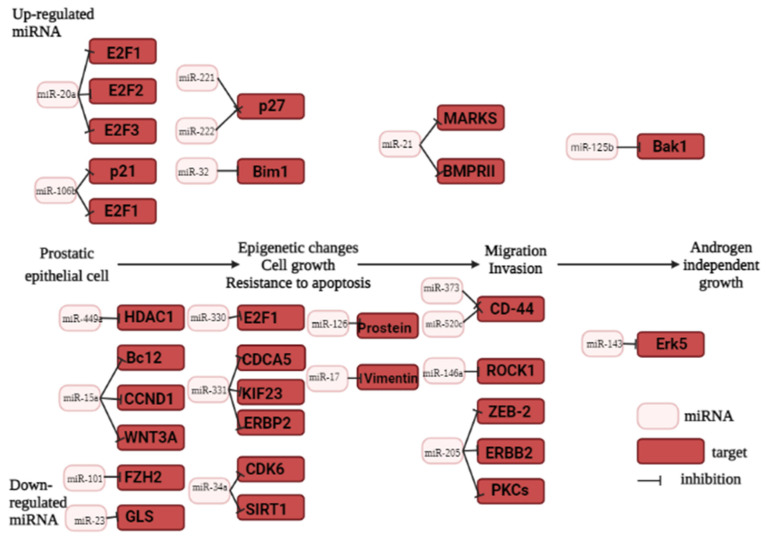
Upregulated and downregulated miRNAs, and their role in the pathogenesis of prostate cancer.

**Figure 3 ijms-24-12797-f003:**
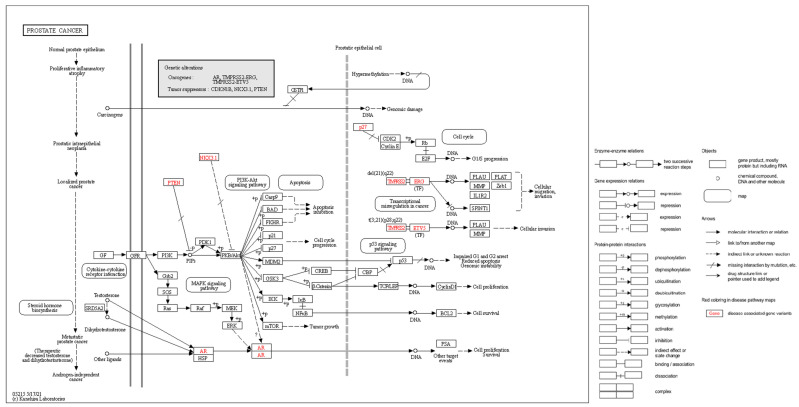
Prostate cancer pathways [112]. This is a schematic presentation of PC pathogenesis, driving factors, connections between genes, and signaling pathways. Key events in carcinogenesis include DNA hypermethylation, triggering the MARK signaling pathway, which leads to cell proliferation and tumor growth. Open questions are marked in red.

**Table 1 ijms-24-12797-t001:** The epidemiology of prostate cancer in 2020 according to the Global Cancer Observatory (GCO) [2].

Region	Incidence			Deaths			5-Year Prevalence (All Ages)	
	Number	% of All Sites	Rank	Number	% of All Sites	Rank	Number	Per 100,000
World	1,414,259	7.3%	3	375,304	3.8%	8	4,956,901	126.13
Europe	473,344	10.8%	3	108,088	5.5%	5	1,873,814	518.11
Northern America	239,574	9.4%	3	37,192	5.3%	5	929,921	-
Latin America and the Caribbean	214,522	14.6%	1	57,415	8.0%	3	709,119	220.48
Asia	371,225	3.9%	8	120,593	2.1%	14	1,176,781	49.59
Africa	93,173	8.4%	3	47,249	6.6%	4	178,197	26.60
Oceania	22,421	8.8%	2	4767	6.9	4	89,069	416.92
Russia Federation	46,454	7.9%	3	14,434	4.6	7	169,221	250.18

**Table 2 ijms-24-12797-t002:** Oncogenic miRNA in prostate cancer.

miRNA	Function	Experimental Models (Cell Lines, Animal Models)	Patient Cohort, Size, Age, and Geographic Location, Groups of Comparison	Reference
miR-18a	Increasing cancer progression	-	160 patients, average age 56.8 ± 12 including stage I, II, and IV presenting to the National Cancer Institute Cairo compared to 50 normal control healthy male individuals	[119]
miR-21	Accelerating tumor invasion and inducing castration resistance	-	170 patients older than 45 years from Zagazig University Hospitals, Egypt compared to 70 healthy men	[125]
miR-32	Inhibition of apoptosis and increased proliferation	transgenic mir-32 mice	-	[30]
miR-106/miR-25	Increasing cancer progression	LNCaP cells PC-3 cells	-	[31]
miR-125b	Increase in cell proliferation and suppression of apoptosis	The human PC cell lines: C4-2 CWR22Rv1 BCa cell lines: T24, TCC-SUP, UMUC3, TCC-5637, and 293T	-	[123]
miR-141	Development of castration resistance	LNCaP cells PC-3 cells	-	[125,126,127]
miR-221/miR-222	Increased cell proliferation, invasion, cell survival	LNCaP, PC3	-	[128]
miR-375	Diagnostics	LNCaP, PC3	-	[127]
miR-650	Reduced expression of the cellular stress response gene 1 (CSR1).	PC3	216 patients aged from 45 through 79 years from Pittsburgh, USA compared to 77 healthy men	[129]
miR-4534	Downregulating the tumor suppressor *PTEN* gene	LNCaP, PC3		[124]

LNCaP cells are androgen-sensitive human prostate adenocarcinoma cells derived from the left supraclavicular lymph node metastasis from a 50-year-old Caucasian male in 1977. PC-3 is a cell line initiated from a bone metastasis of a grade IV prostatic adenocarcinoma from a 62-year-old male. C4-2 is a cell line with epithelial-like morphology that was isolated from a human PC LNCaP cell subcutaneous xenograft tumor of castrated mouse. CWR22Rv1 human prostate carcinoma epithelial cell line derived from a xenograft. BC—bladder cancer. T24 a cell line established from a human urinary bladder cancer patient. TCC-SUP is a cell line isolated from the urinary bladder of a female with grade IV transitional cell carcinoma. UMUC3 is an epithelial-like cell that was isolated from the urinary bladder male of a patient and can be used in cancer research. TCC-5637 is a cell line exhibiting epithelial morphology that was isolated from the urinary bladder of a 68-year-old white male patient with grade II carcinoma. 293T is an immortalized cell line derived from the embryonic human kidney that is transfected with sheared human adenovirus type 5 DNA.

## Data Availability

Not applicable.

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
