# Peer review of "Prostate Cancer: Genetics, Epigenetics and the Need for Immunological Biomarkers"

_ijms, 2023, doi:10.3390/ijms241612797_

Round 1
Reviewer 1 Report
The manuscript titled "Prostate Cancer: Genetics, Epigenetics, and the Role of MicroRNAs as Immunological Biomarkers" by Guzel Rafikova et al. summarized the role of genetics, microRNAs, and immunological biomarkers in prostate cancer. However, I have identified several areas where the manuscript can be improved to ensure a more cohesive and comprehensive discussion of the subject matter.
1. Section 3 and Title:
I agree with your inclusion of microRNAs as a crucial aspect of epigenetic alterations in prostate cancer. Based on the information provided in the abstract and the inclusion of Figure 2 and Table 2, which elucidate the role of microRNAs in prostate cancer, it would be appropriate to add "MicroRNAs" to the title to reflect the content more accurately.
2. Section 5: Methods for Diagnosing Prostate Cancer:
To align the discussion with the need for immunological biomarkers, I suggest reorganizing this section to provide a more direct and focused introduction to immunological biomarkers for prostate cancer diagnosis. Emphasize the limitations of currently used diagnostic tools and explain how immunological biomarkers can address these unsettled needs. Additionally, discuss emerging methods and technologies that show promise in the field of immunological biomarkers for prostate cancer.
3. Relationship between Clinical Characteristics and TAMs:
To enhance the manuscript's comprehensiveness, it would be valuable to discuss the relationship between clinical characteristics and increased tumor-associated macrophages (TAMs) in prostate cancer. Furthermore, I suggest you consider creating a new table that compares TAMs' involvement in prostate cancer with other cancer types, such as lung, breast, ovarian, and colorectal cancers. This addition will provide valuable insights into the broader role of TAMs in various cancer contexts.
4. TAMs and Chemotherapy Response:
Since TAMs may play a role in chemotherapy response, despite contradictory findings and lack of consensus, it is important to discuss this aspect in your manuscript.
I appreciate the opportunity to review your work and look forward to seeing the revised manuscript.
Author Response
The manuscript titled "Prostate Cancer: Genetics, Epigenetics, and the Role of MicroRNAs as Immunological Biomarkers" by Guzel Rafikova et al. summarized the role of genetics, microRNAs, and immunological biomarkers in prostate cancer. However, I have identified several areas where the manuscript can be improved to ensure a more cohesive and comprehensive discussion of the subject matter.
- Section 3 and Title:
I agree with your inclusion of microRNAs as a crucial aspect of epigenetic alterations in prostate cancer. Based on the information provided in the abstract and the inclusion of Figure 2 and Table 2, which elucidate the role of microRNAs in prostate cancer, it would be appropriate to add "MicroRNAs" to the title to reflect the content more accurately.
Answer: We have added other aspects of epigenetics to the article, including DNA methylation and histone modification (lines 344-529). The revised version covers now 3 essential epigenetic levels, and title corresponds to the content.
- Section 5: Methods for Diagnosing Prostate Cancer:
To align the discussion with the need for immunological biomarkers, I suggest reorganizing this section to provide a more direct and focused introduction to immunological biomarkers for prostate cancer diagnosis.
Answer We emphasized the need for immunological biomarkers in prostate cancer diagnosis:
“Search for the immunological biomarkers in prostate cancer has been predominantly focused on regulators of inflammation, mainly cytokines and tumor-infiltrating lymphocytes. Several pro-inflammatory cytokines have emerged as potential biomarkers, as inflammation has been implicated in tumor initiation and progression [180]. For instance, increased levels of IL-8, TNF-α, and MCP-1 (CCL2 )have been associated with worse overall survival in metastatic prostate cancer patients undergoing androgen deprivation therapy [181]. Transforming growth factor-β1 (TGF-β1), involved in cell-mediated immunity, was linked to increased Gleason score, biochemical recurrence after radical prostatectomy, and disease progression [182,183]. Cytokines like IL-8 and stromal cell-derived factor-1 (SDF-1) have also been directly linked to prostate cancer progression [184]. Toll-like receptors (TLRs), responsible for recognition of invading pathogens and metabolic ligands, have been a subject of research, with increased TLR-9 observed in poorly differentiated prostate neoplasms [185]. Dysregulation of TLRs, either up- or down-regulation, has been associated with a high rate of prostate cancer recurrence [186]. Cells of tumor microenvironment have also gained attention as biomarkers in PC. Prostate cancer tumors are often infiltrated by regulatory T cells, PD-1+ cells, and PD-L1+ cells. Elevated PD-1 expression in T cells has been linked to shorter biochemical survival and increased CD8+ lymphocyte density [115,187,188]. In a study exploring genetic and immunological approaches, PLK1 emerged as a potential contributor to prostate cancer progression based on the LnCeVar database. A study utilizing double immunofluorescence tagging of CD163, an M2 macrophage marker, and PLK1 was conducted on human prostate cancer tumor tissue microarrays. The results demonstrated a positive association between PLK1 and CD163 in prostate cancer samples, while no such correlation was observed in benign hyperplasia samples [189]”.
Additionally, discuss emerging methods and technologies that show promise in the field of immunological biomarkers for prostate cancer.
Answer: we have provide following discussion in Conclusions:
“Advanced and precise technologies of spatial transcriptomics and spatial epigenomics, single cell analysis of transcriptome, epigenome and metabolome open the way to identify crucial detrimental immune cell subpopulations as well as key immune biomarkers that can be translated to clinics. However, the application of such technologies requires, in most of cases, prospective patient sample collections and substantial funding”.
- Relationship between Clinical Characteristics and TAMs:
To enhance the manuscript's comprehensiveness, it would be valuable to discuss the increased tumor-associated macrophages (TAMs) in prostate cancer
Answer: We highlighted the links between clinical prostate cancer characteristics and TAMs:
“TAMs play a significant role in prostate cancer pathology. Increased TAM density in prostate cancer demonstrated correlation with higher Gleason score and shorter cancer-specific survival. Additionally, TAMs have been associated with the activation of osteoclast-related pathways, which is particularly relevant considering the propensity of prostate cancer to metastasize to the bones. [217]. Understanding the involvement of TAMs in prostate cancer holds clinical importance for developing targeted therapeutic approaches”.
We make a reference on our recently review, here we provide details about TAMs biology, correlations with PC and outlooks for their clinical values
Front Oncol. 2020 Oct 22;10:566511. doi: 10.3389/fonc.2020.566511. eCollection 2020.
Tumor-Associated Macrophages in Human Breast, Colorectal, Lung, Ovarian and Prostate Cancers
Irina Larionova , Gulnara Tuguzbaeva, Anastasia Ponomaryova , Marina Stakheyeva , Nadezhda Cherdyntseva, Valentin Pavlov , Evgeniy Choinzonov, Julia Kzhyshkowska
Furthermore, I suggest you consider creating a new table that compares TAMs' involvement in prostate cancer with other cancer types, such as lung, breast, ovarian, and colorectal cancers. This addition will provide valuable insights into the broader role of TAMs in various cancer contexts.
Answer: The types of TAMs involved in the carcinogenesis of other types of cancer have been extensively described in a separate publication. doi: 10.3389/fonc.2020.566511
- TAMs and Chemotherapy Response:
Since TAMs may play a role in chemotherapy response, despite contradictory findings and lack of consensus, it is important to discuss this aspect in your manuscript.
Answer: we have added following text about TAMs role in prostate cancer therapy:
“TAMs play a significant role in prostate cancer therapy. TAMs are key components of the tumor microenvironment and have both anti-tumor and anti-inflammatory effects. In prostate cancer, TAMs interact with tumor cells and immune cells, such as T lymphocytes, influencing disease progression [211].
One important function of TAMs in prostate cancer is the regulation of inflammation and tissue remodeling. TAMs express immunosuppressive markers, such as PD-L1, which suppress the activation of effector T cells and facilitate immune evasion by the tumor. TAMs can undergo phenotypic changes during cancer progression, transitioning from an anti-tumor state to an anti-inflammatory state, thereby enhancing immunosuppressive effects and promoting tumor growth [211].
Research suggests that modulating TAM activation may improve the effectiveness of immunotherapy in prostate cancer. Targeting signaling pathways or using specific drugs to modulate TAM activation could enhance immunotherapy efficacy and improve treatment outcomes. However, a deeper understanding of TAM interactions with tumor cells and the tumor microenvironment is needed to develop more effective immunotherapeutic strategies in prostate cancer”.
Reviewer 2 Report
In this article, the authors review the topic of prostate cancer with an emphasis on genetics, epigenetics and the need for immunological biomarkers.
General comment:
The article is highly informative, comprehensive and well written. This reviewer envisons that the article would be a starting point for many researches in the field of prostate cancer.
Specific points:
Figure 3 is of a poor quality and the text is hardly readable.
Epigenetic mechanisms, other than miRNA, are mentioned in the Abstract, however, they are not discussed in the article.
Except for mentioning diagnostic tests in section 5.3, the authors could briefly discuss the gene-expression based prognostic and predictive tests like Decipher, Prolaris and Oncotype DX. This is only a suggestion and I leave it up to the authors if they find it suitable to write about tests other than diagnostic.
Minor points:
Tables 2 and 3: The first raw of the table with the titles of the columns ('miRNA', 'Function', 'Experimental models…') could be emphasized to make it stand out from the rest of the text in the table.
Line 619: It is written 'carcionogenesi'.
In references, it seems that there are some cyrillic letters left. For example 'J. L. Jahn, E. L. Giovannucci, и M. J. Stampfer'.
English language is fine.
Author Response
In this article, the authors review the topic of prostate cancer with an emphasis on genetics, epigenetics and the need for immunological biomarkers.
General comment:
The article is highly informative, comprehensive and well written. This reviewer envisons that the article would be a starting point for many researches in the field of prostate cancer.
Specific points:
Figure 3 is of a poor quality and the text is hardly readable.
Answer: When downsizing an image to A4 format, there is a loss of quality. To mitigate this issue, we will provide the high-resolution image as a separate file.
Epigenetic mechanisms, other than miRNA, are mentioned in the Abstract, however, they are not discussed in the article.
Answer: We have added other aspects of epigenetics to the article, including DNA methylation and histone modification, to provide a more comprehensive description of the impact of key mechanisms of epigenetics on PC carcinogenesis: lines 344-529.
Except for mentioning diagnostic tests in section 5.3, the authors could briefly discuss the gene-expression based prognostic and predictive tests like Decipher, Prolaris and Oncotype DX. This is only a suggestion and I leave it up to the authors if they find it suitable to write about tests other than diagnostic.
Answer: We have added following text about modern predictive tests in the article:
“State-of-the art gene expression-based tests like Oncotype DX Prostate and Prolaris aid in treatment decision-making for prostate biopsies during active surveillance. Decipher is designed for high-risk surgical specimens. Decipher uses 22 genomic classifiers (GCs) with 19 genes to assess genomic risk on a scale of 0 to 1. It analyzes the entire transcriptome and RNA biomarkers associated with cell proliferation, differentiation, motility, and immune respons [202]. Decipher predicts relapse or metastasis after radical prostatectomy in patients with adverse comorbidities [203]. GC-based systems outperform conventional parameters in metastasis prediction [204]. Oncotype DX Prostate predicts adverse events after radical prostatectomy in low-risk patients with 10-20 year life expectancy. It uses 12 genes associated with morphological changes, proliferation, and androgen signaling [205]. Prolaris calculates a risk score based on 31 cell cycle progression genes. It benefits post-biopsy prostatectomy patients and untreated low-risk patients [206]. CCP score predicts risk and mortality in patients treated conservatively [206].”
Minor points:
Tables 2 and 3: The first raw of the table with the titles of the columns ('miRNA', 'Function', 'Experimental models…') could be emphasized to make it stand out from the rest of the text in the table.
Answer: modified as recommended.
Line 619: It is written 'carcionogenesi'.
Answer: corrected to “carcinogenesis”
In references, it seems that there are some cyrillic letters left. For example 'J. L. Jahn, E. L. Giovannucci, и M. J. Stampfer'.
Answer corrected.
Additionally, a portion of the text highlighted in gray has been revised to increase the uniqueness of the article and avoid repetitions.
Round 2
Reviewer 1 Report
Comments to Author:
Thank you for addressing the comments from the first round of review. However, upon careful reading of the latest revision, several errors were identified that need to be corrected.
1. Abbreviations Usage:
In the manuscript, there are instances where the use of abbreviations seems unnecessary or inconsistent. For example:
- Line 629 and Line 642: The abbreviation "PC" is defined at the beginning. Please consider removing the abbreviation in these instances.
- Throughout the manuscript: Several abbreviations like "AUC”, ”CRPC” …
2. Typo and Style Corrections:
- Line 664: "miR-18a-5p expression" should be corrected to "MiR-18a-5p expression" to maintain consistent capitalization.
3. Missing References:
Several references are missing in the manuscript. Please provide the appropriate references for the following lines, among others:
- Line 279
- Line 987
- Line 989
- Line 1092
Please carefully address these issues in your revision to improve the clarity and accuracy of the manuscript.
Author Response
Thank you for the attentive reading and valuable comments on my article. Your feedback was very helpful, and I truly appreciate the time and effort you put into analyzing our work.
- Abbreviations Usage:
In the manuscript, there are instances where the use of abbreviations seems unnecessary or inconsistent. For example:
- Line 629 and Line 642: The abbreviation "PC" is defined at the beginning. Please consider removing the abbreviation in these instances.
- Throughout the manuscript: Several abbreviations like "AUC”, ”CRPC” …
Answer: We have double-checked all the abbreviations and brought them in line with the requirements
- Typo and Style Corrections:
- Line 664: "miR-18a-5p expression" should be corrected to "MiR-18a-5p expression" to maintain consistent capitalization.
Answer: The mistake is corrected
- Missing References:
Several references are missing in the manuscript. Please provide the appropriate references for the following lines, among others:
- Line 279
- Line 987
- Line 989
- Line 1092
Answer: We have verified all references and added them in lines 279,987,989 . Line 1092 reflects the authors' opinion and draws a conclusion for the chapter.
Please carefully address these issues in your revision to improve the clarity and accuracy of the manuscrip
Round 3
Reviewer 1 Report
The authors have successfully fulfilled all of the reviewers' comments.